# Eco-Innovation Maturity Model: A Framework to Support the Evolution of Eco-Innovation Integration in Companies

**Amanda Xavier [1],\*** , **Tatiana Reyes [2]**, **Améziane Aoussat [3]**, **Leandro Luiz [1]** and **Lucas Souza [1]**

[1]   Alberto Luiz Coimbra Institute, Federal University of Rio de Janeiro, Rio de Janeiro 21941-914, Brazil; leandro.luiz@pep.ufrj.br (L.L.); souzalm@gmail.com (L.S.)

[2]   Institut Charles Delaunay, Université de Champagne, Université de Technologie de Troyes—CNRS, 10010 Troyes, France; tatiana.reyes_carrillo@utt.fr

[3]   Laboratoire Conception de Produits et Innovation, Arts et Métiers ParisTech, 75013 Paris, France; ameziane.aoussat@ensam.eu

\*   Correspondence: amandaxavier86@gmail.com; Tel.: +55-21-99-774-5678

**Abstract:** The urgency for sustainable changes in product performance and around the processes of the different organizational areas highlights the potential of eco-innovation as a management strategy. However, this holistic approach to eco-innovation is still a challenge for businesses. In this sense, this research was structured on two central premises, which justify the central problem of the study. The first is that companies face organizational barriers to the implementation of eco-innovation in a holistic manner. The second is that companies face operational barriers to the implementation and global integration of eco-innovation, such as lack of models, methods, and support tools. This operational barrier mainly includes the lack of maturity approaches and prescriptive methods of eco-innovation evaluation. In order to overcome these barriers, this research proposes an Eco-innovation Maturity Model (Eco-Mi), a framework to support eco-innovation integration and evolution of organizational maturity. Based on a review of the literature, it was possible to develop a first version of the Eco-innovation Maturity Model, consisting of the eco-innovation maturity levels and a guide to eco-innovation practices. The model has been improved through expert evaluation with the use of the Delphi Method, which contributed to increase its validity and reliability. The results confirm the research hypothesis and, therefore, the validity of the Eco-Mi model as support for the integration and evolution of eco-innovation in organizations and as a reference for the field of knowledge.

**Keywords:** eco-innovation maturity model; eco-innovation practices; eco-innovation management; eco-innovation assessment

## 1. Introduction

The urgency for progressive sustainable changes in product performance and around processes in different organizational areas highlights the potential of eco-innovation as a management strategy. Eco-innovations contribute to a sustainable environment through the development of ecological improvements [1–4], and may comprise not only eco-friendly products, processes, and services, but also organizational management systems that are sensitive to environmental concerns and system innovations [5,6]. Eco-innovation provides extensive contributions to the achievement of long-term sustainability in order to integrate the environmental dimension through the whole process, not only at the eco-design stage. It includes a change of the functionalities required in new products and a change of its business model [7]. This way, not only environmental impacts but also social impacts are

reduced [8]. This perspective involves broad strategic vision, implying holistic changes throughout the organization, which also include its stakeholders [9]. This means that sustainability is seen not only as an operational excellence exercise, but as an innovation that requires different organizational dynamics [10]. By bringing environmental aspects into discussions, eco-innovations can affect and transform the innovation system in order to create sustainable processes [11].

This need for progressive sustainable changes in the performance of products and around the processes of different organizational areas highlights the potential of eco-innovation as a management strategy. It is necessary to understand how we can encourage corporate eco-innovation, so that it significantly changes the way companies operate to ensure greater sustainability [12]. In this way, it is necessary to work collaboratively across the organization, involving not only the three levels of the company (strategic, tactical, and operational), but also all its stakeholders. In general, it is a culturewide change within the organization for sustainability [4,13].

However, the combination of several factors represents an increasing challenge for companies and their managers to promote changes in their practices in a manner to equalize economic, environmental, and social responsibilities towards sustainable development [14]. In this sense, this research was structured on two central premises, which justify the central problem of the study. The first is that companies face organizational barriers to the implementation of eco-innovation with respect to strategy, structure, resources, culture, and immediate vision issues. The second is that companies face operational barriers to the implementation and global integration of eco-innovation, such as lack of models, methods, and support tools. This operational barrier mainly includes the lack of maturity approaches and prescriptive methods of evaluating eco-innovation. These gaps regarding operational and organizational barriers are presented in Table 1.

**Table 1.** Organizational and operational barriers.

| **Organizational Barriers** | |
| --- | --- |
| Lack of strategies or reactive and timely vision for sustainable development | Galvão (2014) [14]; Kemp and Pearson (2007) [15]; Morrish et al. (2011) [16]; Kuratko et al. (2014) [17]; Xavier et al. (2015) [8]; Fernando and Wah (2017) [18]; Del Río et al. (2010) [7] |
| Conflicts between structure and strategy, and inefficient management | Masera (2004) [19]; Hrebiniak (2005) [20]; Mello and Nascimento (2005) [21]; Tachizawa and Andrade (2008) [22]; Aubry et al. (2008) [23] |
| Lack of appropriate resources and initiatives | Epstein and Roy (2001) [24]; Masera (2004) [19]; Hrebiniak (2005) [20]; Mello and Nascimento (2005) [21]; Tachizawa and Andrade (20080 [22]; Jabbour and Santos (2008) [25]; Del Río et al. (2010) [7] |
| Lack of a culture (values and organizational climate) of innovation and sustainability | Schein (2004) [26]; Colbert et al. (2008) [27]; Green et al. (2008) [28]; Baker et al. (2014) [29]; Jin et al. (2019) [30]; García-Machado and Martínez-Ávila (2019) [31] |
| Short-term vision and learning process focused on solving specific problems | Hellstrom (2007) [32]; Carrillo-Hermosilla et al. (2010) [33]; Albuquerque (2011) [34]; Hofstra and Huisingh (2014) [35] |
| **Operational barriers** | |
| Lack of eco-innovation models, methods and tools in the literature | Samet (2010) [36]; Hautamäki (2010) [37]; Roscoe et al. (2016) [38]; Blaise (2014) [39]; Xavier et al. (2017) [40] |
| Lack of models focused on the holistic and systemic integration of eco-innovation | ISO (2011) [41]; Gouvinhas et al. (2016) [13]; Xavier et al. (2017) [40] |
| Lack of eco-innovation maturity models | Ormazabal et al. (2016) [42]; Xavier et al. (2019) [43] |
| Lack of prescriptive methods for assessing the maturity of eco-innovation | Jabbour (2010) [44]; Torres (2016) [45]; Pöppelbuß and Röglinger (2011) [46]; Ormazabal and Sarriegi (2012) [47]; Xavier et al. (2017) [40]; Xavier et al. (2019) [43]; Munodawafa and Johl (2019) [48] |

These gaps emphasize the importance of the development of instruments that represent an organized environment to advise organizations on the management, enabling the consecutive improvement of the phenomena and the effectiveness of the management. To fill this gap, innovation and sustainability management methods with a broad strategic vision are needed [49], which significantly

changes the way companies operate [12], integrating the environmental dimension through the entire innovation process [39].

A concept that can provide holistic support for a transformation project for management and that allows an evaluation to be performed is the maturity model [45]. Maturity modeling is a generic approach that describes the development of an organization over time progression through ideal levels for a final state [50]. It describes a process by which the organization can develop some improvement, such as a set of skills or practices, resulting in a more mature organization [51]. The main value of a maturity assessment is to capture the company's perception towards the current situation so that it can improve itself [52]. In this way, maturity models are tools used to evaluate organizational elements and select appropriate actions that take these elements to higher levels of maturity [53].

Although the constructs of evolutionary stages in management are useful, the literature is still incipient: it offers merely descriptive classification [43,44], and lacks on providing a detailed definition of each phase and a way to advance from one phase to the next on the organizational evolution [43,54].

This research deals with the factors that make up the management process of innovation and sustainability, through the global understanding of its elements and determinants, as well as their interactions. In this way, it will be possible to systematize practices in eco-innovation management, in order to aid performance evaluation and business decision-making, and to facilitate the process of holistic improvements and changes in its evolution. This research intends to answer the following research question: How to systematize eco-innovation practices in order to support eco-innovation integration and evolution of organizational maturity? Based on relevant references from the field of knowledge, which present organizational and operational barriers to integration of eco-innovation in organizations, the research hypothesis is as follows: An eco-innovation maturity model can provide framework of eco-innovation practices and a guide to support eco-innovation and evolution in organizations.

The main objective of this research is to propose an Eco-innovation Maturity Model, in order to provide a framework to support eco-innovation integration and evolution of organizational maturity. The holistic approach and the prescriptive method for assessing and improving the organization can be highlighted as the originality of this research. From the general objective, specific objectives are developed:

- to provide a framework of eco-innovation practices in order to promote the diffusion and growth of the field of knowledge;
- to develop and characterize eco-innovation maturity levels;
- to develop a systematic and prescriptive method to evaluate eco-innovation practices in companies.

As a scientific contribution, the proposition of a framework for eco-innovation management practices can be highlighted, as well as the identification and characterization of the levels of maturity of eco-innovation. As a practical contribution, it will be possible to propose to the industrial environment: a simple and practical guide to the selection and global integration of eco-innovation practices in organizations; an instrument for evaluating organizational performance in eco-innovation; a case study to disseminate better industrial practices.

This paper is divided into six sections, the first section being devoted to introduction. In the next sections, the following topics are addressed: Section 2 discusses the background theory; Section 3 presents the research method used in this research; Section 4 presents the development of the Eco-innovation Maturity Model; Section 5 presents the case study; and Section 6 presents the conclusions and suggestions for future researches.

## 2. Background Theory

Eco-efficiency and corporate social responsibility practices alone, while important, are not enough to deliver the holistic changes necessary to achieve long-term social and environmental sustainability [12]. In other words, creating a sustainable corporate image is not just a question of

developing certain "sustainable products"; it also requires that all management procedures within the company be based on a different philosophy and a strategic vision focused on sustainability. This new vision must permeate all sectors and departments of the company [13]. Sustainability must be incorporated as part of the way the company conducts its business instead of considering it as something "beyond" its general business practices and procedures [38]. Innovative management should encourage the continued development of new management models and methods to effectively manage innovation processes, as well as to motivate and stimulate staff towards creativity and innovation, strategic agility, and the ability to quickly capture the possibilities of environmental action [37].

It is assumed that organizations that implement strategic decisions integrated to the environmental aspect minimize their risk, gain competitive advantages, as well as face cost reduction and increase profits at the medium and long terms [7,22,55–58]. This means that when the economic, environmental, and social aspects of innovation are dealt and inserted in the company's strategy, its innovative potential is maximized, because the proactive behavior systematically changes the organization in its goals, values, and culture, and increases the innovative, economic, and sustainable results [8]. Eco-innovations may involve a combination of elements pertaining to all those dimensions, which play a significant role in understanding the multi-faceted nature of eco-innovations and their diversity. When addressed together, they form a comprehensive framework for the analysis of eco-innovation [7]. In this way, manufacturers must be proactive towards environmental issues and link their operations beyond the economic rewards, to consider environmental and social impacts in their processes [59].

From this holistic approach, an important option for companies is to choose among different ways to operationalize strategies that allow them to benefit from more open and sustainable approaches [60] and different ways of opening their innovative process [61]. Therefore, in addition to the design context and development of new products, eco-innovation is also studied in the field of management and innovation strategy, as well as in management, strategy, and environmental policy. This is due to the fact that sustainability is seen not only as an operational excellence exercise, but as an innovation that requires different organizational dynamics [10]. Besides, in order to reach the sustainable goals, innovation is an important mechanism driven by the continuous need for quality improvement and by policy measures and regulation [55,62].

Therefore, it is extremely important to understand how sustainability is integrated in the process of innovation management. Although the processes follow the same steps in the process of eco-innovation, unlike conventional, sustainability is an integrated objective in corporate policy and, therefore, in the process success factor, having different methods and success indicators [63,64]. However, it is not always clear how a sustainable process is organized within a company. Although many authors see innovation as a key factor in sustainability, little attention is paid to how firms can find and develop eco-innovations [38]. The same authors argue that sustainability should be taken as part of the way an organization conducts its business, rather than something "beyond" its general business practices and procedures. In this sense, innovative management stimulates companies to continually develop and test new models and methods of management, in order to manage the innovation processes effectively, as well as to motivate and stimulate their personnel towards creativity and innovation, strategic agility, and the ability to grasp the possibilities offered by the environmental action quickly [37].

Since innovation is the core business process, companies need to find a way to organize and manage the innovation process in order to ensure its growth [65]. Well-structured processes are not enough for innovation to occur. The execution of these processes will always be in charge of people and it is impossible to ignore the relevance of factors related to the way these people relate to one another, with the projects and the organization, the configurations that permeate the company, and the ways in which the different functions interact. A context that supports and promotes innovative activity is needed [66].

Several authors have addressed the dimensions of innovation management with different approaches, such as "corporate conditions" for innovation [67], "contextual factors" of innovation [65], or "enabling context" for creative work [68]. These proposals have been made to organize and

classify innovation management into categories, allowing better understanding and its evaluation and measurement from fundamental points. This process includes the formulation of strategies and the use of the organizational structure as a way to group and coordinate resources to achieve the objectives [65]. The three dimensions mentioned—strategy, structure and resources—are proposed in different classifications as the main ones in the construction of an innovative environment [65,66,69–74]. Some approaches present the structure divided into other categories, such as structure, leadership, and process [69]; structure and process [70]; structure and organization [71]; organization and process [73]. Similarly, the resources dimension can be analyzed from the perspectives: people, investment and relationship [69]; people and rewards [70]; resources and technology [73,75]. In addition to these three dimensions, one dimension that appears in several classifications is culture, which is analyzed in terms of a generic category of culture [66,69], in the aspect of external relationship [66], and in relation to the organizational learning [73,76,77].

Therefore, based on the dimensions proposed by [65] and the analysis carried out by comparing the different dimensions proposed in the literature, the essential elements of the context of innovation management are treated and distributed in this research through four interdependent dimensions: Strategy, Structure, Resources, and Culture.

In face of the considerable number of different dimensions of innovation management, this combination of factors, both procedural and contextual, has made it increasingly challenging for the company to make changes in its practices, in the sense of responding to the challenges of sustainability [14]. This statement is aligned with the fact that the vast majority of companies have a reactive profile, neither have effective actions on environmental strategy nor intention of applying them [3].

In this context, organizations need guidance on how to systematically apply their efforts in order to achieve environmental objectives and maintain continuous improvement in the environmental performance of products and processes [41]. Despite this, there are few eco-innovation models, methods, and tools [36,40] and, even those that do exist, pay little attention to how companies develop and integrate these eco-innovations [38,40]. In addition, although there are methods and tools available for the improvement of environmental management, companies are at different stages of maturity, and there are few classifications that deeply explain how a company can reach and overcome more evolved levels of maturity [42]. Although the constructs of evolutionary stages in management are useful, the value of a maturity model is in its processes and causal analyses, which help organizations to improve and advance in the maturity scales [42].

There is a growing increase in the proposal for maturity models [42,54,78–83]. Likewise, it can be noticed that the development of models is aimed at the development of sustainable products [13,84–86]. Among these models, the framework for corporate self-assessment of organizational maturity in sustainability [13] and the EcoM2 model [84] can be highlighted: the former takes into account the holistic and strategic aspects, in addition to the operational ones; the latter presents a prescriptive maturity model, with an application method that encompasses not only a diagnosis of the profile in ecodesign, but a whole process of implementation of improvement.

Only two models can be specifically cited in the theme of sustainable innovation: methodology for analyzing the maturity of sustainable innovations [87]; and model of evolutionary stages in sustainable innovation management [88]. However, these models have some limitations. Despite their simple and practical structure, the level of detail of the first model [87] is superficial and its list of practices is strongly limited to the GRI (Global Reporting Initiative), in the sense that it offers no evolution in the indicators or goal to be achieved. This aspect makes it difficult to compare performance between companies. In addition, there is no assessment of organizational performance within the maturity levels, that is, analysis of the percentage of compliance with the criteria for each stage, which reduces the practicality of the improvement and the evolution process by the company. Considering Delai's maturity model [88], it has a detailed level of development, but its characterization is purely descriptive.

In addition, there is no maturity assessment method that can be applied in the business environment, depending exclusively on its replication by new researchers in the academic field.

In view of this analysis, there is a scarcity of mature models for eco-innovation; an absence of models with prescriptive characteristics; lack of method to support both the diagnosis phase and the implementation of improvements. Therefore, the effective management of eco-innovation requires the consideration of maturity models that integrate the best innovation and sustainability practices in different areas and organizational levels, contemplating holistic changes and systematic efforts to guide not only the assessment of organizational maturity, but also the holistic improvements and changes for its evolution.

## 3. Materials and Methods

The first stage of the research is exploratory, which aims to investigate poorly understood phenomena, and to explore the problem or situation in order to provide criteria and understanding [53,89]. Therefore, the research foresees the accomplishment of systematic literature review for the construction of the conceptual theoretical structure [90], through the survey and analysis of studies on innovation and sustainability. The first activity of this stage has been a systematic literature review about eco-innovation models [40]. Systematic review is a specific methodology used to map and synthesize a specific theme, providing a rigorous and reliable basis of literature review [91,92]. Therefore, this systematic review has raised the main organizational and operational barriers to integrate eco-innovation in companies. The second activity of this stage has been a systematic literature review concerning maturity models of eco-innovation and related areas, in order to understand the main characteristics and elements of these maturity models, especially in relation to the construction of the levels of evolution and the form of application in the companies [40]. With the systematic review, it was possible to locate existing studies, select, analyze, and synthesize data in such a way that it allows reasonably clear conclusions to be reached about what is and is not known [93]. From this, it was possible to develop the research hypothesis (Section 1) and the conceptual model. The conceptual model illustrates the four dimensions of the eco-innovation maturity model and their interrelations, as shown in Figure 1.

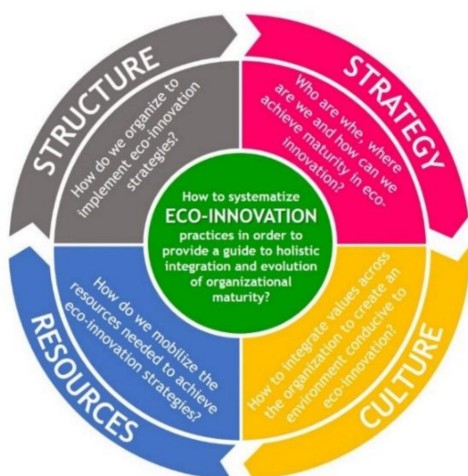

**Figure 1.** Conceptual model.

The four dimensions of innovation management were developed based on the three dimensions (strategy, structure, and resources) proposed by Tidd and Bessant [65], called "contextual factors", including "culture" dimensions proposed by other authors [66,69,73,76,77], as presented in Section 2. The conceptual model guided the literature review regarding studies that addressed sustainability in the four dimensions of organizational innovation management. This was the first step of the second stage.

For the structured literature review, Scopus and ISI Web of Science databases were used, which have a wide scope in the search for academic literature [94], and Google Scholar database was also used, for broad repository of theses, dissertations, and book chapters. As keywords, equivalent terms for eco-innovation used by the main authors of the field of knowledge were selected: sustainable innovation, green innovation, environmental innovation. These terms are analyzed by different authors, who conclude that these dimensions cover much of the literature on eco-innovation [3,8,95–97]. In addition to these, the keywords strategy, structure, resources, and culture were used, which are related to the four dimensions of the conceptual model. This results in the following research strings: "eco-innovation" or "sustainable innovation" or "green innovation" or "environmental innovation" and "strategy" or "structure" or "resources" or "culture", being applied to the titles, abstracts, and keywords of the bases.

With the bibliographic search, about 2100 studies were obtained, including articles, theses, book chapters, and other publications. The abstracts of these studies were analyzed considering the inclusion criteria, and 560 studies were selected. Each of the 560 studies were analyzed through a second filter. During the reading process and evaluation of these studies, those that did not meet the inclusion criteria were excluded, even if they contained some of the keywords or search strings. Then, studies that present guidelines and/or business practices related to the management of organizational innovation (considering the four dimensions) and sustainability (economic, social, and environmental) were selected. From this, 35 studies that presented eco-innovation practices related to strategic dimension, 42 related to structure, 50 related to resources, and 37 related to culture were selected. This activity resulted in the development of the conceptual theoretical framework, which was the basis for the construction of the maturity model. This activity resulted in the development of the conceptual theoretical framework, which was the basis for the construction of the maturity model.

Facing this, the second stage contemplates the development of the Eco-Mi model, which presents a normative/prescriptive characteristic according to its interest in developing strategies, checklists, actions, and other types of tools that aim to improve the results available in the existing literature and to implement the concept in practice [89,98]. To do so, five major activities were carried out, based on the stages of theoretical development of EcoM2 maturity model [99], as shown in Figure 2. At the end of this stage, an Eco-Innovation Maturity Model (Eco-Mi) is proposed, consisting of a Guide to Eco-innovation Practices, Eco-innovation Maturity Levels, and an Application Method of Eco-Mi model. This stage also contemplates exploratory research with specialist researchers and consultants in the area of innovation management and environmental management, which allows for the evaluation, validation, and identification of fundamental aspects that was not approached only with the theoretical reference. The evaluation of the Eco-Mi model was completed through the Delphi method, which seeks a consensus of opinion about the Model regarding adherence to the concepts and robustness, this way contributing to the possibility of improvements and necessary adjustments [100]. To do so, we selected seven specialists in research and reference laboratories—who have scientific publications in indexed journals—and consultants in the area of innovation and environmental management—with a minimum period of 5 years of experience. The specialists represent academic scientific knowledge and experience in consulting services in diversified industries, according to guidance for the Delphi method [101].

Through the validation of the Eco-Mi model by the Delphi Method, the Eco-Mi Assessment Tool (instrument of the application method) was tested in a case study at a benchmark company in innovation and sustainability. The case study method allows to identify the critical factors of a contemporary phenomenon within a real life context [102]. Therefore, with the objective of generating knowledge for practical application, this is an applied research and its results aim at the solution of a specific problem found in the reality [89,103]. To do so, a qualitative approach was used to understand the environment through observation and interpretation of the objects of study [89]. In qualitative research, the possibilities of generalization result from the adequacy between the analyzed phenomenon and the theory under development, and not necessarily from the number of cases studied [104,105]. Single

case studies allow for a more precise understanding of the circumstances in which the phenomena occurred and, therefore, tend to be more reliable [105,106].

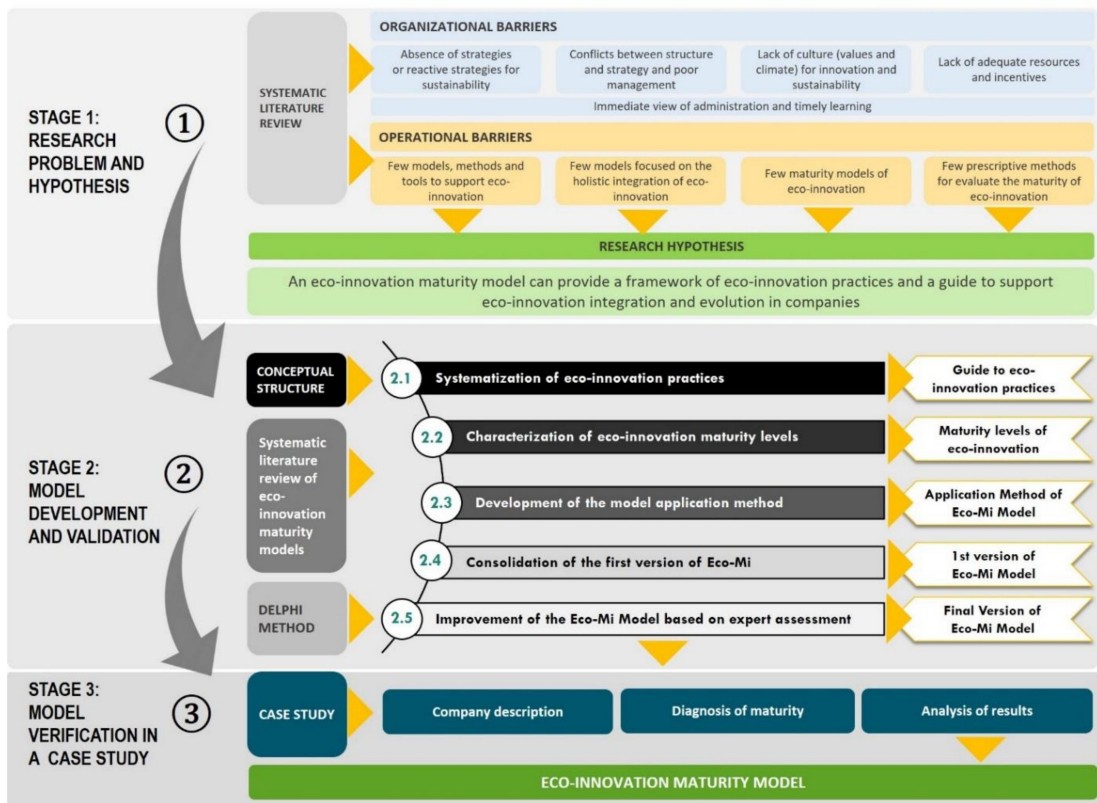

**Figure 2.** Major activities to the development of the Eco-innovation Maturity Model (Eco-Mi).

In order to specify the type of organization to be approached and the type of data to be collected [104], the following question was established: What is the level of maturity in eco-innovation of a company internationally recognized as one of the most innovative and sustainable in the world? The criteria used to choose the company were [102] company that develops products and/or technologies; company in some national ranking of innovation/sustainability; company certified by a globally recognized entity, which establishes standards for managing social and environmental responsibility.

Thus, a Brazilian petrochemical company, internationally recognized as one of the most innovative and sustainable in the world, was selected. The company is the largest producer of thermoplastic resins in the Americas, the world leader in the production of biopolymers, and the largest producer of polypropylene in the United States. As instruments of data collection, interviews, document collection and analysis, and non-participant observation in the studied organization were used, considered as valid instruments for the case study [102]. The Manager of the product development process of the studied petrochemical company was submitted to four interviews, guided through a structured questionnaire (Eco-Mi evaluation instrument) containing general procedures and rules for conducting and completing them. The case study will be described in detail in Section 4.

After completing the case study, it was possible to formalize what is done by the company, proposing a detailed diagnosis of the level of organizational maturity of eco-innovation. From the analysis of the results, it was possible to propose specific improvements for the company and also new approaches and strategic tools of monitoring and control for the integration of eco-innovation practices. The activities of the stages are described in more detail in the following topic (Section 4) regarding model development.

## 4. Development of Eco-Innovation Maturity Model

The Eco-innovation Maturity Model is a framework to support the evolution of eco-innovation integration in companies. The model is composed by three main elements: (1) Guide to Eco-innovation Practices; (2) Maturity Levels of eco-innovation; (3) Application Method, composed by an assessment tool and proposals to improve the integration of practices (Figure 3). Each of these elements will be explained in detail in the following topics.

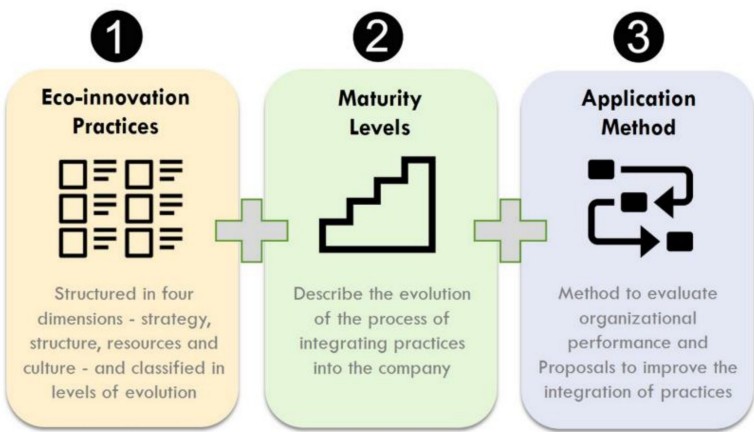

**Figure 3.** The three elements of the Eco-Mi model.

### 4.1. Systematization of the Guide to Eco-Innovation Practices

Eco-innovation practices are defined, for the purposes of this research, as eco-innovation activities that integrate sustainability aspects (economic, social, and environmental) in all dimensions that contextualize innovation management in organizations (strategy, structure, resources, and culture).

A total of 142 practices from the literature on eco-innovation were selected (according to the procedure presented in Section 3), classified according to the four dimensions of the conceptual model: Strategy, Structure, Resources, and Culture. Then, they were systematized into subdimensions, according to their theoretical standards and categories [107]. In order to systematize eco-innovation practices, research questions were proposed for each of the four dimensions. These questions guide the classification of identified practices and facilitate systematization in subdimensions. Thus, in order to provide a guide to holistic integration and evolution of organizational maturity, we selected eco-innovation practices with the aim of responding to the following questions:

- Strategy: Who are we, where are we, and how can we achieve maturity in eco-innovation? (Subdimensions: Diagnosis, Formulation, Control);
- Structure: How do we organize to implement eco-innovation strategies? (Subdimensions: Process, Leadership, Organizational Architecture);
- Resources: How do we mobilize the resources needed to achieve eco-innovation strategies? (Subdimensions: Human Resources, Financial Resources, Infrastructure, Relational Competences);
- Culture: How to integrate values across the organization to create an environment conducive to eco-innovation? (Subdimensions: Eco-innovative culture, Organizational Climate, Organizational Learning).

The classification activity also includes a codification of the practices in order to facilitate the presentation and evaluation [108] in the Delphi method. Figure 4 illustrates some of the eco-innovation practices and coding developed. As a result, a first version of the Guide to eco-innovation practices was developed, with 142 eco-innovation practices to be evaluated in the Policy Delphi method. The practices selected are related to the holistic approach of eco-innovation, that is, they incorporate competences of different organizational dimensions. This is evident, since the practices are systematized into four

global dimensions, which deal with the managerial totality of an organization. However, although practices are classified in dimensions, they are interrelated, since the holistic approach foresees the interrelation between different processes of different competences. In addition, the practices are generic and therefore adaptable to any type of company that develops products and innovation.

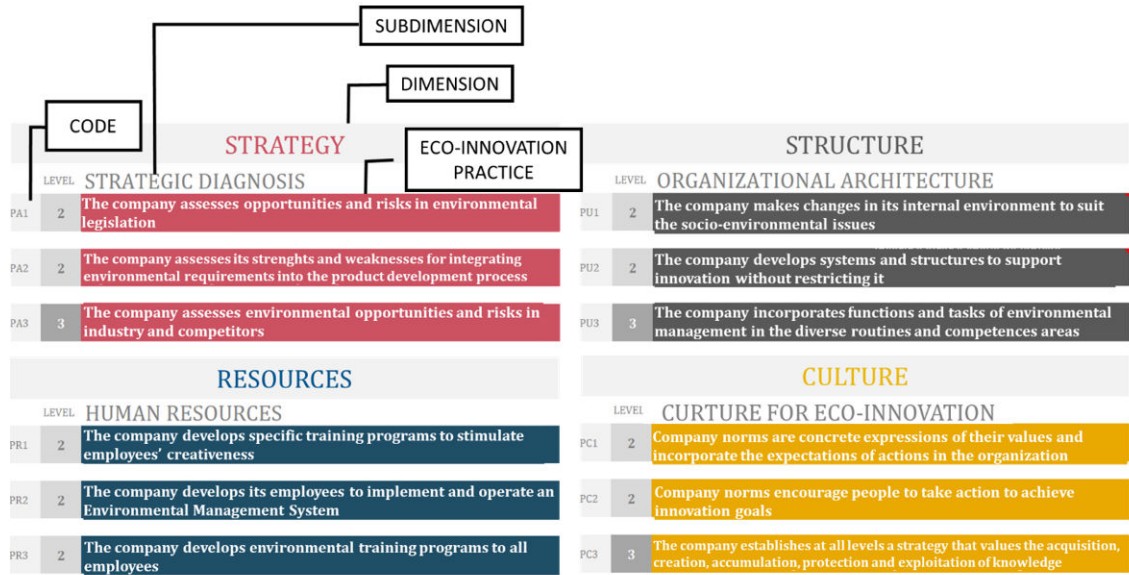

**Figure 4.** Coding of the practices of the Eco-Mi model.

### 4.2. Characterization of Eco-Innovation Maturity Levels

Through a systematic review and analysis of the literature maturity models in eco-innovation and related areas [43], the levels of eco-innovation maturity were developed according to the four dimensions of the conceptual model. Maturity models usually present their evolution path described by a limited number of maturity levels (usually 4 to 6), ordered sequentially, and characterized by certain requirements to be met [50]. Characterization includes detailed description of each level, based on the primordial practices of each dimension [84]. The result of this activity is the first version of the Eco-Innovation Maturity Levels, which describes a recommendation on the stages to be followed in order to achieve a holistic integration of eco-innovation practices in the organization.

To this end, the first step was to analyze and compare the maturity levels of existing eco-innovation and eco-design models. Four maturity models were used. Two models are of ecodesign: the framework proposed by Gouvinhas [13]; "EcoM2" proposed by Pigosso [99]. The other two models are eco-innovation models: Methodology for maturity analysis of sustainable innovations [87]; Model of evolutionary stages in sustainable innovation management [88].

It is possible to verify that the models have an evolutionary pattern, considering four main levels: (1) first level, where the company does not apply sustainable practices; (2) initial and reactive level, where practices are applied on time and/or without formalization; (3) proactive level, focused on eco-efficiency, in which the company already has maturity, and sustainable practices are formally applied to the product development process; (4) proactive strategic level, where the company has globally/holistically integrated sustainability and also works with its external partners. There are also some differentiated practices that justify an even more advanced level of maturity in eco-innovation, which can be seen in the 'Type 6' model of Gouvinhas [13] and is briefly incorporated into the 'Strategic' level of the model proposed by Delai [88]. These practices relate to the systemic characteristic of eco-innovation, present in mature and highly innovative companies that are concerned with educating their value chain, from first suppliers to end customers. This can be done directly (trainings, forums, events) or indirectly (through cooperatives, associations, other partners), in order to work in a collaborative network for sustainability.

Considering the comparative analysis of these maturity models selected in the literature and the analysis of 142 systematized eco-innovation practices, five levels of eco-innovation maturity were proposed. These levels represent the evolution of organizational maturity, from the non-application of eco-innovative practices to the application and holistic integration of eco-innovation practices in the organization. Therefore, the levels demonstrate the ways that the company can follow in order to improve its eco-innovative performance through the integration of practices in the company. The five levels of evolution were structured according to the four dimensions of the model, as shown on Figure 5.

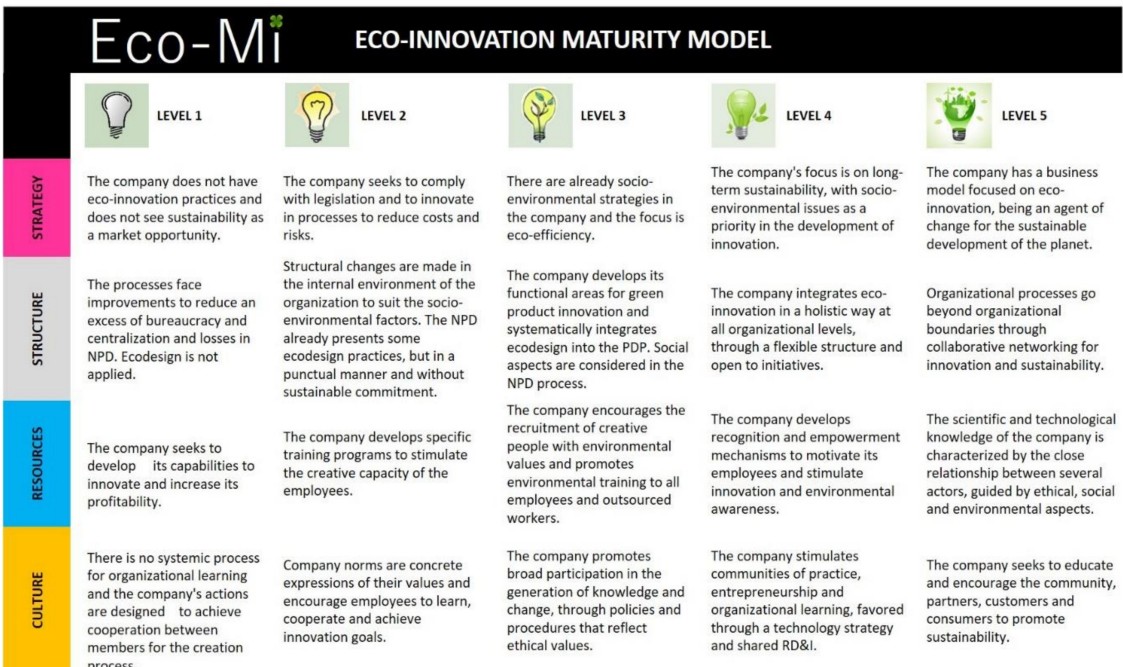

**Figure 5.** Eco-innovation maturity model.

### 4.3. Development of the Application Method

In order to achieve the development of a prescribed approach, the characteristics of each stage of the maturity model and the logical relationship between successive stages need to be explained [46,109]. Therefore, it is necessary to propose evaluation criteria for each level of maturity available [110], so that it can present a high level of verifiability through precise, concise, and clear descriptions to discriminate levels [111]. As well as the criteria, the evaluation methodology must also be verifiable, through the development of a procedure that guides users in the evaluation stages [46] and in the evaluation scales [112].

This development of the model application method includes the development of a systematic and prescriptive method of evaluating organizational performance in eco-innovation. The method also includes qualitative analysis of the results, since the information of each company should be analyzed in isolation, considering aspects of the sector and company profile. It should be noted that the application method was developed in a way that can be easily replicated by companies and researchers. For this, the method also includes propositions of approaches and tools of monitoring and strategic control, to support the improvement of the integration of eco-innovation practices in companies.

Therefore, the application method is composed of an organizational performance assessment method—with a five-level capability model and an Eco-Mi assessment tool (Section 4.3.1), and by propositions of approaches and tools of control and strategic monitoring (Section 4.3.2), such as Steering Committee and the Balanced Scorecard tool (BSC Eco-Mi). The application of the method is composed of three main steps, according to Figure 6:

1.　Evaluation of the maturity of eco-innovation: through interviews with the organization and the use of the Eco-Mi Assessment Instrument;
2.　Analysis and improvement propositions: through the qualitative and quantitative analysis of the results of the evaluation and with the aid of a specialist in the area;
3.　Control and monitoring of results: through translation of improvement propositions into measurable and controllable objectives as well as control and monitoring of results, which can be done through a steering committee and through the support of the Balanced Scorecard (BSC Eco-Mi) tool.

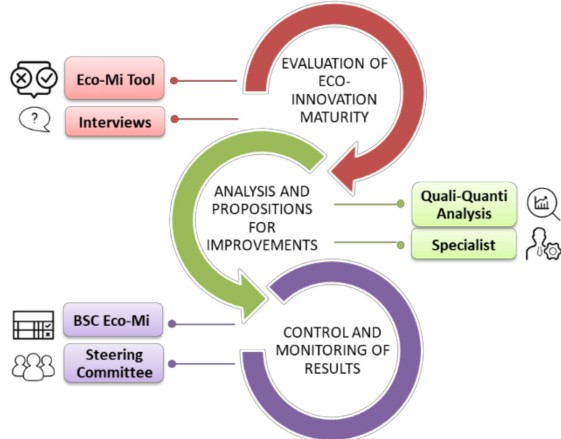

**Figure 6.** Stages of application of the Eco-Mi method.

### 4.3.1. Eco-Mi Evaluation Method

From the characterization of maturity levels and the systematization of eco-innovation practices by dimension, subdimension, and level, an evaluation methodology was established through assessment scales [85,112], criteria for level discrimination [110], and a procedure model to guide users in completing the organizational assessment tool [46]. The assessment scales were called capability levels, based on the CMMI model (Capability Maturity Model Integration), created from(and to meet the limitations of) the CMM model [113]; and based on the capability model adapted by Pigosso [99].

Considering that the Eco-Mi model has a strategic perspective that takes into account the intentional factor of integrating eco-innovation through a holistic approach (in all sectors of the organization, in an interrelated way), it was adopted as the basis for the development of capabilities for the set of eco-innovation components proposed by Iñigo and Albareda [114]: operational, collaborative, organizational, instrumental, and holistic. These components can be understood as adaptive dynamics, showing that they are able to adapt, learn, and generate new structures, rules, and behaviors at different interrelated levels of the company and its systemic environments. By analyzing these components, which are very well suited to the dimensions of the Eco-Mi model, and the capacitance models discussed above, the five levels of the Eco-Mi model are proposed, as shown in Figure 7.

The five proposed levels present the holistic evolution of the integration of eco-innovation practices in organizations, starting with the absence of the practice of eco-innovation or its incomplete application to a strategic and systemic integration of the practice in all the organizational sectors. Through this growing scale of assessment, companies can express the degree of agreement with the level of capability, that is, the level of integration of each eco-innovation practice in the company.

In addition to the evaluation scales, the evaluation criteria should be defined to discriminate each of the five levels of eco-innovation evolution. It was adopted as a criterion for setting the company at a certain level of evolution that 90% of the practices of the level (and of the previous ones) have a capability equal to or greater than the level "Operational" (C3) (Figure 8). Thus, for the company to fit

Level 4, for example, it is necessary that 90% of the practices of Level 2, Level 3, and Level 4 have a capability greater than or equal to C3.

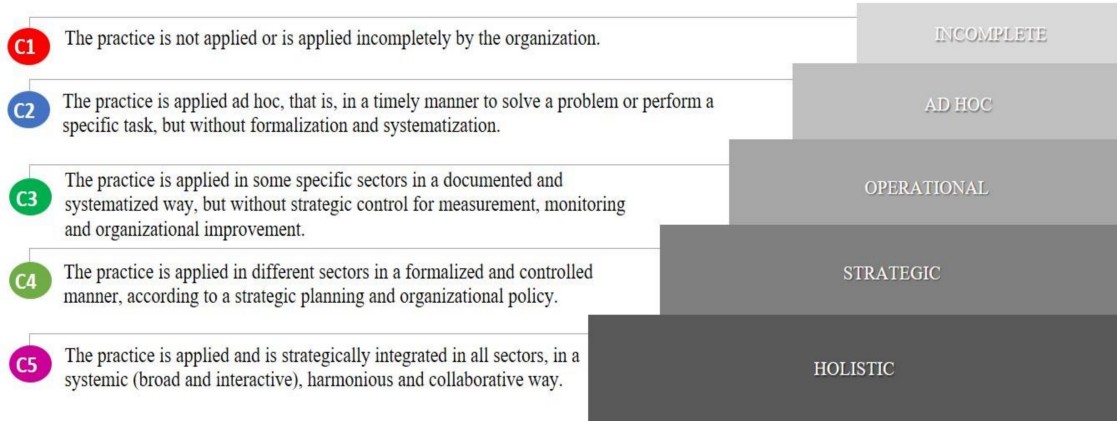

**Figure 7.** Capability levels of the Eco-Mi model.

|         | Level 1 | Level 2 | Level 3 | Level 4 | Level 5 |
|---------|---------|---------|---------|---------|---------|
| Level 2 | <90% practices >=C3 | >=90% practices >=C3 | >=90% practices >=C3 | >=90% practices >=C3 | >=90% practices >=C3 |
| Level 3 |         |         | >=90% practices >=C3 | >=90% practices >=C3 | >=90% practices >=C3 |
| Level 4 |         |         |         | >=90% practices >=C3 | >=90% practices >=C3 |
| Level 5 |         |         |         |         | >=90% practices >=C3 |

**Figure 8.** Matrix of maturity levels and discrimination criteria.

Finally, evaluation methodologies need to present a procedure model that guides users to maturity assessments, elaborating evaluation steps, their interactions, and also on how to respond to evaluation scales [46,111,115]. To this end, an instrument for evaluating organizational performance in eco-innovation was developed, with the purpose of educating respondents and collecting information in a simple and didactic way, maximizing the performance of evaluation results.

The Eco-Mi evaluation tool was developed in Microsoft Office Excel through a spreadsheet that instructs, collects information, and generates a quantitative partial result (Figure 9). The Excel worksheet can be requested by e-mail to the author. The evaluation method includes the collection of information through structured interviews in the organization for the diagnosis of maturity. Interviews should preferably be face-to-face, so that the evaluator/specialist can present the Eco-Mi maturity model and the Assessment Tool, and instruct the respondent throughout the completion and evaluation. It is suggested to interview the person responsible for the process of innovation and/or product and technology development, as well as those responsible for strategic activities, human and socio-cultural resources. However, in mature eco-innovation companies, it is believed that an innovation manager will have sufficient holistic knowledge to respond to the entire assessment tool, and may consult third-party colleagues on issues of doubt. It should be noted that the method was developed in a way that can be easily replicated by companies and researchers. Companies can therefore make use of the Eco-Mi method through self-assessment—whether by internal expert or not, as other academic researchers may replicate the method in other studies.

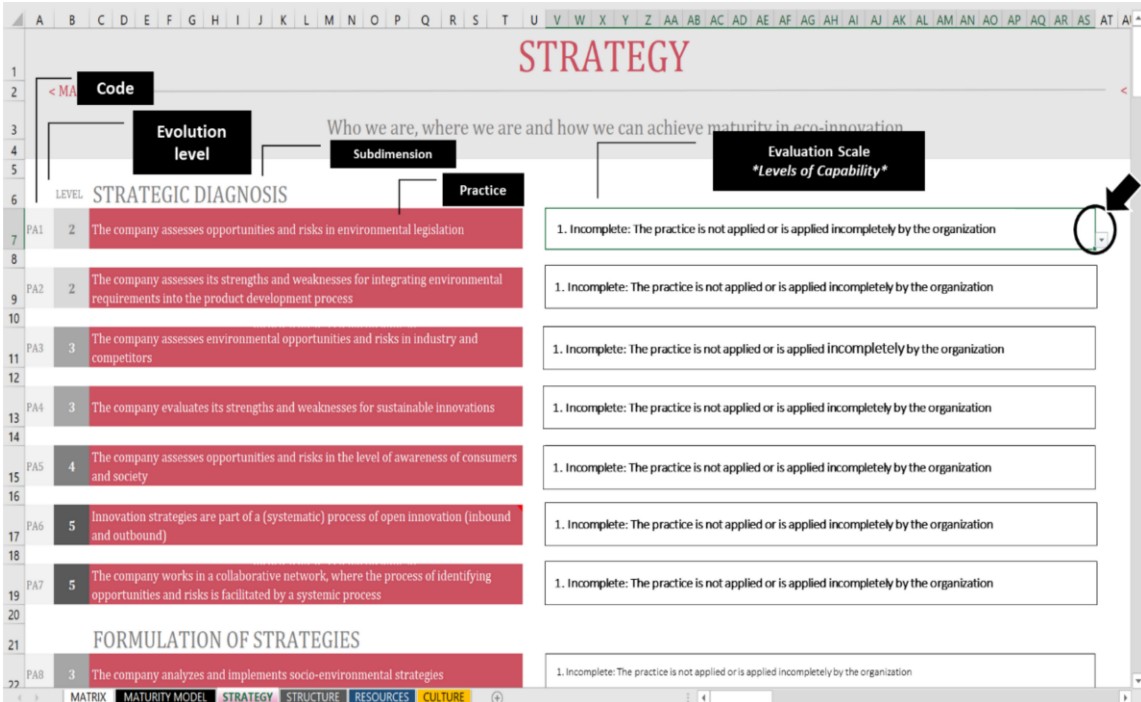

**Figure 9.** Tab Strategy—instrument for assessing the maturity of eco-innovation.

### 4.3.2. Propositions of Approaches and Tools for Improvement

A model that provides prescriptive actions to improve maturity should have the resources available to track interventions longitudinally. This capability will further support model standardization and global acceptance [115]. The decision methodology should also provide advice on how to implement and adapt improvement measures [46]. Thus, in order to support the improved integration of eco-innovation practices and organizational performance evolution, the Eco-Mi Application Method encompasses two proposals to monitoring and strategic control:

1.  Steering Committee: Steering Committees are a group of people chosen to guide and monitor projects in large organizations as part of project governance [116]. The performance of the Steering Committee, inserted in the Eco-Mi application method, shapes a series of improvement objectives around the results of organizational performance in eco-innovation (Section 4.3.1), guiding the company towards the improvement of eco-innovation practices. Discussions increase transparency and trust between participants, generating opportunities to common positions and shared goals. In addition, it promotes coordination of the objectives set and their transition in results, through intensive monitoring and control [117]. The Committee may discuss issues such as what the differentials and limitations of the company are; what the main practices that limited the company's maturity were; if the limitations have been due to absence or low level of capability; in what dimensions the company is stronger/weaker; what the standard capability of the company is; what limits the level of capability; how to improve.

2.  Balanced Scorecard tool (BSC Eco-Mi): The Balanced Scorecard (BSC) methodology [118] can be used to translate sustainability strategies into the business reality of Steering Committees, or simply of organizations. It facilitates the visualization of the results achieved, generating a better understanding and alignment of all involved [119]. Given this context, the BSC presents itself as a bridging tool to support the improvement strategies raised by the Eco-Mi evaluation method, diagnosing which practices should be integrated or better applied in the company. The use of the BSC for sustainability can also be carried out in a specific way, in order to keep the issues addressed separate from the global BSC [118]. As a way of facilitating the development of

the BSC strategic map, it is proposed the use of the four dimensions that constitute the Eco-Mi model, as perspectives of the strategic map. This works as an example of a methodology based on the BSC for measuring environmental performance [120]. The points of improvement, referring to one of the four dimensions of the model, translate into measurable and controllable objectives in the BSC Eco-Mi strategic map. For each strategic objective, goals and performance indicators should be aligned, thus creating the Balanced Scorecard [119]. In addition, an action plan may be proposed to support the objectives described [121]. To this end, the BSC Eco-Mi instrument was developed using a spreadsheet in Microsoft Office Excel to longitudinally support the conversion of improvement points into measurable objectives. The Excel spreadsheet can be requested by email to the author.

### 4.4. Improvement of the Eco-Mi Model Based on Expert Assessment

The Eco-Mi model was evaluated using the Policy Delphi method, which searches the opinions of experts about the practices and maturity levels proposed and adherence to the concepts and robustness, allowing for the possibility of improvements and adjustments necessary [101,122]. The main activities for the development of Delphi are as follows: select the specialists; develop the evaluation questionnaire; plan and conduct the Delphi sequences/rounds; analyze the results and systematize comments and suggestions for improvement; develop the final version of the Eco-Mi model [123]. Seven specialists, four academic researchers, and four industry consultants were intentionally selected. The experts represent academic scientific knowledge and practical experience through consultancies in diversified industries, as shown in Table 2 below. In addition, they have complementary knowledge, covering the entire context of the research object, and are from distinct geographical locations, which increases the diversification of learning [100]. The experts were contacted by email and invited to contribute to the evaluation of the Eco-Mi model.

**Table 2.** Experts for the evaluation of Eco-Mi.

| | |
|---|---|
| **Expert 1** | Researcher in Innovation for Sustainability<br>Lappeenranta University of Technology (London, England) |
| **Expert 2** | Teacher and Researcher in Ecodesign<br>Université de technologie de Troyes UTT (Troyes, France) |
| **Expert 3** | Specialist and Analyst in Sustainability and Governance<br>SGS Brasil (São Paulo, Brazil) |
| **Expert 4** | Specialist and Consultant in Strategy and Innovation<br>EloGroup Consulting (Rio de Janeiro, Brazil) |
| **Expert 5** | Teacher and Researcher in Ecodesign<br>Federal University of Rio Grande do Norte (Natal, Brazil) |
| **Expert 6** | Specialist and Consultant in Environmental Management<br>Roguier Consulting (Rio de Janeiro, Brazil) |
| **Expert 7** | Professor and Head of LCPI (Laboratory of Product Design and Innovation)<br>Arts et Métiers (Ensam Paristech) (Paris, France) |

The most efficient way to structure a Delphi questionnaire is feeding the questions using the existing research material itself [100]. In this way, the structure of the Delphi questionnaire was based on the consolidated version of the Eco-Mi model and was structured in a Microsoft Excel spreadsheet. This research considers all points of view, and from the first sequence, suggestions for changes were made referring to the comments of each expert. Due to this, a minimum of two sequences were predicted to evaluate the Eco-Mi model [124]. The purpose of the second sequence was to present all suggestions for improvements proposed in the first sequence for a further evaluation of the experts in order to reach a consensus of approvals of those changes [101]. After receiving the answers of the second round, a new analysis and compilation of the results was performed, seeking to verify the consensus according to the established minimum criterion (70%) [125,126]. It is worth mentioning that

in the majority of cases the consensus was 86% (six of seven specialists). In this way, with the second round, it was possible to reach consensus in almost all practices based on suggestions and comments.

Of the 142 practices, 76 were approved without alterations, 33 were modified, 31 were excluded, and two were directed to other dimensions. As suggested by experts, three practices were added. Thus, 112 eco-innovation practices were validated. Regarding the levels, 74 were approved directly and 38 were modified according to the experts' suggestions. The list of validated practices and levels can be seen in Table 3 below. The main modifications suggested were the tangibilization of practices, that is, changes in order to facilitate the verifiability of the practice in organizations. Other changes suggested the inclusion of comments that would facilitate or illustrate the understanding of some of the concepts mentioned. Finally, there were changes with the intention of improving the sentence for better understanding. The excluded practices, in turn, were related to the ambiguity that some had with others, or to the non-direct relation to eco-innovation.

**Table 3.** Validated practices and levels.

|  | Strategy | Structure | Resources | Culture | Total |
|---|---|---|---|---|---|
| Practices (1st version) | 24 | 35 | 48 | 35 | 142 |
| Practices validated | 23 | 25 | 37 | 27 | 112 |
|  | **Level 2** | **Level 3** | **Level 4** | **Level 5** | **Total** |
| Practices validated | 23 | 37 | 34 | 18 | 112 |

All experts' comments in the second round were satisfactory and demonstrate alignment with expectations of the results generated. The comments confirm the reliability of the model and robustness of both the content of the model and the method developed for evaluation, which demonstrated practicality and simplicity for filling and managing the information. From the results analyzed, it was possible to develop the final version of the Eco-Mi model. The Appendix A presents the final version of the guide to eco-innovation practices, systematized by dimensions and maturity levels.

## 5. Case Study

This topic presents the results of the verification of the Eco-Mi Maturity Model (Eco-Mi) in a Brazilian multinational reference in innovation and sustainability. The domain in which the Eco-Mi model can be applied is composed by companies that present a structured process of innovation for the development of new products and/or technologies, and that aim at the integration of eco-innovation practices. In this way, a Brazilian petrochemical company, internationally recognized as one of the most innovative and sustainable in the world, was selected. For that, a detailed description and evaluation of the maturity of the case was carried out, as well as the analysis of the results. The case study followed the steps of the Eco-Mi application method, to evaluate organizational performance in eco-innovation and analysis and propositions of improvements, and followed the assumptions of the case study method proposed by the theoretical reference.

### 5.1. Petrochemical Industry

The petrochemical industry is a sector of the chemical industry that uses oil products and natural gas as raw material. Petrochemicals is the largest sector in the chemical industry and has high revenues [127]. The petrochemical industry is of great importance worldwide, with oil being one of the largest sources of energy used today, with a share of approximately 32% in world energy consumption. In addition, it still has a prominent participation during the last decades [128,129]. According to data from the Brazilian Chemical Industry Association [130], the estimated worldwide turnover is U$ 4.2 billion, which was responsible for approximately 4.8% of the world GDP in 2018 (Gross Domestic Product). In Brazil, it is estimated that the participation of the chemical sector in GDP reached 2.41% in 2017. The Brazilian chemical industry holds the fourth largest sectorial participation in the country and occupies the ninth position in the world ranking of the sector.

For 2020, it is estimated that the participation of the so-called green chemistry will be at least 10% in the set of petrochemical products in Brazil, strengthening the development of a renewable base industry. The opportunities for developing a renewable-based chemical industry translate into a demand for research and development of both new products and advanced processes [131].

Compared to the other sectors, the petrochemical industry has strong barriers to the entry of new players, in addition to being capital intensive, that is, dependent on a large volume of investments (fixed assets) to maintain its position [132]. The petrochemical industry is strongly characterized by research, development, and innovation (R&D) [133]. Some factors directly affect production costs and strongly contribute to the performance of companies operating in this industry, namely: technology, for process productivity and production scale; flexibility, to incorporate new advances that can contribute to improvements in productivity; location, geographical distance from consumer markets and sources of raw material; ability to store products in the favorable cycle phases to use it in the unfavorable phases; and replacement, where it is possible, of current raw materials with alternative raw materials [134].

The importance of this sector for the world economy can also be verified by the strong connection with other large industrial sectors and by the dependence that several other industries in the chain have in relation to petrochemical products. For all the factors illustrated, it can be asserted that no other sector of activity prescinds the chemical sector today, which makes the presence of this industry strategic in developed and developing economies [131,135].

## 5.2. Summary of the Company Description

The company is the largest producer of the thermoplastic resins sector (polyethylene, polypropylene, and polyvinyl chloride) in the Americas, the world leader in the production of biopolymers and the largest producer of polypropylene in the United States. With an annual production of 16 million tons of material, including chemical and basic petrochemicals, the company is also the world's largest producer of biopolymers, with an annual production capacity of 200,000 tons of Green Plastic—100% renewable polyethylene origin.

With approximately R$ 50 billion revenue and R$ 9 billion operating profit, the company has clients in more than 70 countries, assisted by 16 regional offices located worldwide. With an average of 3.5% of revenues invested in product research and development (R&D), the company has 965 patents registered in Brazil and abroad. In 2015, the company invested R$280 million in innovation and technology. There are approximately 300 professionals dedicated to R&D, two Centers of Innovation and Technology located in Brazil and the United States, 23 laboratories, and seven pilot plants, committed to the continuous development of the petrochemical industry and the plastic chain.

The innovation put the Brazilian Petrochemicals as the world's largest manufacturer of biopolymers on an industrial scale, with one of the main highlights being Green Polyethylene, 100% renewable raw material plastic and created from its own technology after three years of R&D investments. In 2015, petrochemicals ranked fourth in the ranking of the 100 most innovative companies in the country, published by the Brazilian newspaper Valor Econômico in partnership with consultancy Strategy &, which for more than a decade has published a global ranking of innovation.

The achievement came one year after being voted as one of the 50 most innovative companies in the world by the American magazine Fast Company. In addition, the company also stands out in its sustainable initiatives. In 2013, it was elected the most sustainable company in Brazil, according to Guia Exame de Sustentabilidade (a Brazilian sustainable guide). Since 2010, it has been included among the 20 Brazilian model companies of this sustainable guide. In 2018, the company integrated the supplier list "Water A" and, for the second time, the supplier list "Climate A", from the CDP (Carbon Disclosure Program) Supply Chain ranking, which assesses the companies that better engage their suppliers.

## 5.3. Diagnosis of Maturity

Based on the quantitative partial result proposed by the Eco-Mi Assessment Instrument and the qualitative analysis of the information collected (through interviews, internal documents and secondary

sources), the eco-innovation maturity level of the petrochemical company can be diagnosed. With a percentage of holistic capability over 90%, eco-innovation practices are applied, integrated, and constantly improved in the petrochemical company studied. No practices with lower than C3 (operational) capability were identified. Moreover, the dimensions are well aligned, containing only one or two practices that are not applied holistically, that is, with systemic integration throughout the organization.

The qualitative analysis of the results validates the diagnosis proposed by the Eco-Mi Assessment Instrument, which allows for the verification that the petrochemical company has a maturity level 5. Its business model is geared towards eco-innovation and the company is an agent of change for the sustainable development of the planet. Organizational processes go beyond organizational boundaries through collaborative networking for innovation and sustainability. Scientific and technological knowledge is characterized by the close relationship between several actors, guided by ethical, social, and environmental aspects. In addition, the company seeks to educate and encourage the community, its partners, customers, and consumers to promote sustainability.

*5.4. Improvement Proposals*

From the analysis of the results and diagnosis of the eco-innovation maturity of the petrochemical company studied, it is possible to diagnose the points of improvement, that is, which practices should be integrated or better applied in the company to improve its performance and eco-innovative potential. These points of improvement, referring to one of the four dimensions of the model, could be translated into measurable and controllable objectives. To this end, suggestions and advice on approaches and tools for monitoring and strategic control for improvement were also proposed. Figure 10 below summarizes the practices of each dimension of eco-innovation that may have improved capability and goals and improvement propositions.

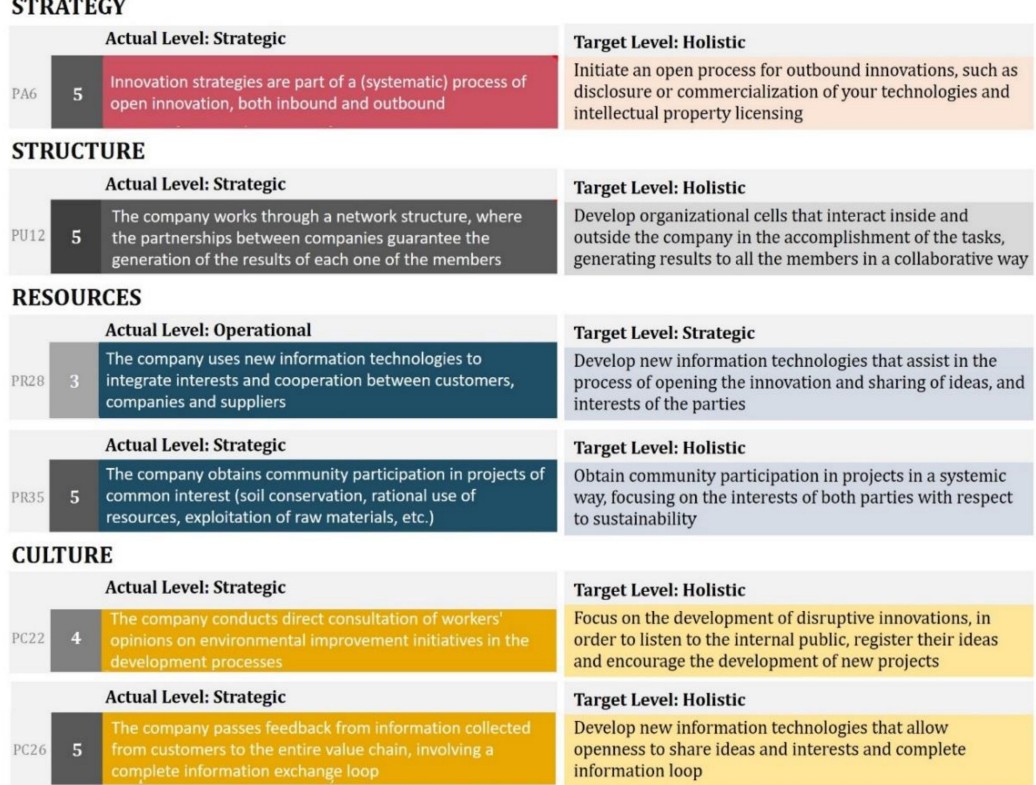

**Figure 10.** Improvement proposals.

To control and monitor improvement outcomes, such propositions can be translated into measurable and controllable goals. This translation activity, which will shape improvement objectives around the results analyzed, can be done through a steering committee. The propositions outlined here should therefore be discussed, directed, and deployed in goals and indicators for control and monitoring. One possible methodology to assist the performance of the Steering Committees is the BSC Eco-Mi instrument. An Action Plan through the BSC Eco-Mi instrument was proposed to the petrochemical company studied, focused on the proposals to improve the eco-innovation practices.

*5.5. Discussion of Results*

The qualitative analysis of the results also showed some eco-innovation practices that confirm the validity of the elements and dimensions of the Eco-Mi model and its potential of use. In addition, the validity of the model, especially the Eco-Mi application method, can be confirmed through an evaluation according to the company's perception after its application, based on 10 criteria (utility, consistency, comprehensiveness, clarity, etc.). The following topics therefore present the pillars and influence factors in innovation and sustainability, which leverage eco-innovation in the petrochemical company and demonstrate strong alignment with the elements and dimensions of the Eco-Mi model; and the results of the evaluation of the model by the company, evidencing its reliability and constructive validity.

5.5.1. Pillars and Factors Influencing Eco-Innovation

The qualitative analysis of the results of the case study shows some relevant findings that confirm the validity of the elements and dimensions of the Eco-Mi model and its potential for use. The first finding refers to the holistic approach, since the aspects of innovation and sustainability were confirmed as a result of systemic collaboration. The holistic approach predicts the interrelationship among different processes of distinct competences [13,40,41], being the basis for the Eco-Mi model, since it has four interdependent dimensions. Therefore, the results of the case study presented a strong alignment with the four dimensions of the Eco-Mi model: strategy, structure, resources, and culture. This alignment is evidenced by the interdependence between practices of different dimensions that presented compatible levels of capability.

In this way, as already presented in greater detail in the theoretical basis, eco-innovation can be explored within a more advanced transformation of the system [114], where work becomes collaborative among all levels of the organization [136,137]. This holistic aspect can also be verified through the ease of performing the diagnosis proposed by the Eco-Mi Application Method. The method proposes as primary contact with the person responsible for the process of innovation and/or development of products and technology, assuming that this professional has enough knowledge to respond to the Eco-Mi Evaluation Instrument. This assumption is based on the premise that mature eco-innovation companies have holistic management and, in this case, knowledge is shared across all sectors of the organization in a strategic and systemic way. Senior managers must be centrally involved in processes that define entrepreneurship for innovation [17,138]. In this sense, it was possible to verify that the holistic approach is a central characteristic of the petrochemical company, since the manager of innovation and knowledge management fully and totally owned the Eco-Mi Evaluation Instrument.

It is also worth noting that the holistic component stimulates the expansion of a new innovation paradigm for human development and socio-environmental ecosystems [114]. This perspective can be confirmed by the fact that sustainability is systematically integrated in all areas of the petrochemical company, aligned with development and innovation goals. To this end, the principle of decentralization and delegation is used to guarantee autonomy for each business unit, through the principle of leader-led hierarchy [139,140]. This structure supports empowerment and empowerment programs for devising and delivering creative and challenging tasks, initiatives that contribute to innovation pointed out in the literature [65,141–143]. The development and diffusion of innovations can also be associated with various forms of dialogue and interactions with stakeholders, which are important sources of information and learning [144,145].

Against this, a second finding can be highlighted: the focus on people is central to the holistic approach. Therefore, human resource management influences all components of the organization. It produces the talent required by the organization's strategy and structure, generating the necessary skills to implement the goals and objectives [70], and is seen as the center of organizational sustainability [25]. Thus, the focus on people is verified as one of the main characteristics of the petrochemical company, which has as its principles: trust in people, in their capability and desire to evolve, and self-development through education. The participation in education and training programs must be continuous for all employees, from upper-management to the bottom of the pyramid [25,146–148]. The holistic approach and focus on people can be considered as pillars of eco-innovation, and confirm the validity of the elements and dimensions of the Eco-Mi model and its potential for use. These pillars were shown as levers for the integration of eco-innovation in the petrochemical company, characterized by four main factors, associated to the four dimensions of the Eco-Mi model, as presented in detail below: sustainability as a success factor; delegation and decentralization; empowerment; direct interactions with stakeholders.

Sustainability as a Factor of Success (Strategy Dimension)

Sustainability is integrated into the innovation management process as an explicit strategic objective, being a success factor in the process of developing new products [64]. The principles of petrochemicals are based on socio-environmental values, such as the principle of customer satisfaction, in order to serve it with quality, productivity and, above all, economic, social, and environmental responsibility. Furthermore, the principle of reinvestment of results focuses on the creation of job opportunities and the development of communities [148]. Economic, environmental, and social factors must be fully incorporated into strategic planning perspectives and into the company's innovation development processes [16,72,149]. In this way, the company values are associated with the sustainability strategy, based on increasingly sustainable processes and resources; increasingly sustainable product portfolio; solutions for society to have an increasingly sustainable life.

In view of this, the company leads industry initiatives to strengthen the environmental attributes of plastic, through Life Cycle Assessment (LCA) studies. Ecodesign practices are systematically incorporated into all phases of product and process development, including social criteria, initiated by both top management and operational levels [99,150]. The results can be verified in the main Sustainable Stock Indexes, of which the petrochemical company includes: Dow Jones Sustainability Emerging Markets Index; BM & FBovespa's Corporate Sustainability Index (ISE); Efficient Carbon Index (ICO2) of BM & FBOVESPA. The Sustainable Stock Indices leverage companies to raise funds in the financial market [151,152], and the sustainability reports may provide information regarding their social and environmental performance [153–155].

These achievements confirm the company's commitment to the best global sustainability practices in the industrial sector by auditing sustainability-related data with the same treatment and rigor as financial data. Sustainability is therefore an explicit strategy and objective [72,118,156,157], and an integrated success factor of its innovation process in the development of new products [72,149], making it easier for the eco-innovation process to be constantly evaluated through specific indicators.

Delegation and Decentralization (Structure Dimension)

The principle of decentralization and delegation is used to guarantee autonomy for each business unit, in petrochemicals considered as a small business. For this, the structure must allow the leaders a decentralized action and with delegation of responsibilities [17,29,130,158]. In addition, the principle should influence the projects regarding the formation of culture and people management in the organization [158].

Based on a decentralized performance, the petrochemical company employs the concept of planned delegation through the leader-led relationship and the Program of Action (PA) as a tool for disseminating strategy, culture, and rewards. The PA unites people to their responsibilities of action, communication, and strategic goals, and gathers the evaluation metrics of each member. Therefore, it is

fundamental to communicate and guarantee the achievement of goals at all levels of the company [159]. It is through PA that the company guarantees a frequent channel of communication between leader and leader. It should be emphasized that the petrochemical structure is based on the performance of the leader within the hierarchy. As a result, the company trains, empowers, and stimulates the entrepreneurial behavior of its personnel, through the valorization of the ideation and recognition of the efforts [160,161]. As a result, innovative initiatives also emerge from lower organizational levels, with appropriate recognition and direction by senior managers. All this empowerment increases the commitment of the members, improves decisions, innovation, and environmental awareness, as pointed out by the literature [17,158].

Empowerment and Training (Resource Dimension)

Empowerment is perceived through the practices of encouraging the autonomy and independence of members, for the ideation and accomplishment of creative and challenging tasks [130,162,163]. All the members have an individual development plan, which specifies the education actions, such as training and training courses. There is a specific planning for the different career moments of the members, and the process is understood as the succession of challenges with increasing complexity. In addition, the company develops reward programs for innovative teams, that promote ideas that contribute to improvement and eco-innovation [70,142,143,164–166]. The strategy of valorization and recognition of efforts is verifiable at all levels of the company, and is supported by the Program of Action. PAs, in turn, function strategically as mechanisms for the empowerment of their employees, encouraging them to take on programs every time, leading to more challenges, and promoting their growth, career development, and perpetuity of business.

Direct Interactions with Stakeholders (Culture Dimension)

The scientific and technological knowledge is characterized by the close relationship between several actors, whose interactions allow the company to form a knowledge base and technological qualification [167]. In this sense, socio-environmental responsibility is associated with the various forms of dialogue and interactions with stakeholders, which constitute important sources of information and learning, favored through shared PD&I [14]. It is possible to perceive that the company works in a network of cooperation to promote the development and the diffusion of the innovations. Partnerships with companies, universities, and laboratories are fundamental for R&D, especially in chemicals and polymers. Nonetheless, considering only the Brazilian context, the petrochemical company establishes partnerships with 15 universities and research institutes with the objective of finding new solutions for the petrochemical and renewable products market.

The company promotes various forms of dialogue and interactions with stakeholders, including new information technologies, in order to facilitate the management of communication [15]. The petrochemical company's focus is on direct interactions with the various stakeholders. The main means of engagement with stakeholders, especially suppliers and customers, are face-to-face meetings, in addition to site visits and participation in collective bodies. Direct and interpersonal contact helps stimulate the inclusion of socio-environmental concerns in suppliers, processes, and procedures [13,15,168]. This is fundamental for the company, which has the socio-environmental performance as a key criterion for the choice of the company's suppliers. In addition, in order to ensure an action connected to the reality and needs of surrounding communities [146,148], the company has local leaders and dedicated institutional relations teams that maintain an open and constant dialogue with these communities, not only monitoring concerns and potential negative impacts but also working on projects focused on human development. In this way, the company's leadership is recognized in the external environment as a reference in eco-innovation.

Finally, the opening of the company to the realization of the present research can be highlighted, with full support and interest in the aid to the scientific knowledge for learning and improvement. This supports an analytical approach to how companies get involved and shape their eco-innovation,

transforming the dynamics between agents, business units, and external partners. The reason for this, mainly, is because the configuration of eco-innovation evolves and is characterized by the complex nature of organizational interactions, which allows companies to evolve continuously, thanks to the dynamics of adaptability [114].

5.5.2. Application Evaluation of the Eco-Mi Model in the Petrochemical Company

The practices highlighted in the case studied show the alignment between the elements and dimensions of the Eco-Mi model and among other theoretical references, previously discussed. This alignment confirms the validity of the Eco-Mi model: the content of the Good Practice Guide aligned with the maturity levels. Additionally, above all, the potential of using the Application Method, both in industries and in academia. Its generalization is, primarily, due to the in-depth analysis of the case, in the sense that it allows for a more precise understanding of the circumstances in which eco-innovation practices occur. In addition, its representativeness is due to the adequacy of the phenomenon analyzed in the company with the theory under development [105]. In this perspective, the contribution of this case study has as its main role to precisely show specific issues, generating possibilities of insights for both theory and business practice.

In addition, the model was evaluated by the company through an evaluation questionnaire sent to the respondents. This questionnaire was developed to evaluate the Eco-Mi model, according to the company's perception after its application. The questionnaire has a series of criteria related to utility, consistency, scope, comprehensiveness, precision, depth, simplicity, clarity, coherence, and instrumentality. For each criterion, four levels of response are proposed: very satisfactory, satisfactory, needs improvements, unsatisfactory. Finally, there is a space for comments and suggestions.

As a result, all the proposed criteria were satisfactorily evaluated by the petrochemical company, what represents consistency of the model with the industrial practice and robustness of the application method. The diagnosis of maturity was in accordance with what the company hoped for, since its strategy is directed towards innovation and sustainability and is, therefore, internationally recognized for the results achieved in this direction. Improvement proposals were also received in a satisfactory manner, reaffirming some issues that can be reflected for improvements in the medium and long term. Although the company already possesses its own approaches and tools for monitoring and strategic control, it recognized the value of new proposals and the interest in learning and improvement in this issue. New discussions were then proposed for the maturing of the perspectives generated and the creation of a collaborative channel for the generation of knowledge in favor of eco-innovation. The Section 6 discusses in greater detail the results achieved, against the proposed objectives, the theoretical and practical contributions of the Eco-Mi model, as well as the limits of the research and the suggestions for future work.

## 6. Conclusions

The urgency for progressive sustainable changes in product performance and around processes in different organizational areas highlights the potential of eco-innovation as a management strategy. Eco-innovation provides extensive contributions to the achievement of long-term sustainability in order to integrate the environmental dimension through the whole process, not only at the eco-design stage. This perspective involves broad strategic vision, implying holistic changes throughout the organization, which also involve its stakeholders. Despite the evolution of sustainable initiatives, this holistic approach to eco-innovation is still a challenge for businesses. In this sense, this research was structured on two central premises, which justify the central problem of the study. The first is that companies face organizational barriers to the implementation of eco-innovation with respect to strategy, structure, resources, and culture. The second is that companies face operational barriers to the implementation and global integration of eco-innovation, such as lack of models, methods, and support tools. There are few mature approaches and prescriptive methods of evaluating eco-innovation.

In order to break these barriers, the aim of this research was to systematize eco-innovation practices through a maturity model, in order to provide a guide to holistic integration and evolution of

organizational maturity. The results and contributions that constitute the originality of this research stand out: (1) guide to eco-innovation practices; (2) maturity levels of eco-innovation; (3) method to evaluate the organizational performance of eco-innovation. The model was improved through expert evaluation using the Delphi Method, which allowed to increase its validity and reliability.

Accordingly, a Brazilian petrochemical company, internationally recognized as one of the most innovative and sustainable in the world, was selected to verify the Eco-Mi model. The evaluation of the eco-innovation performance, through the qualitative-quantitative analysis proposed by the Eco-Mi Application Method, provided a diagnosis of the maturity of the petrochemical company, with a maturity level of 5. The qualitative analysis of the results of the case study showed that a holistic approach and focus on people can be considered as pillars of eco-innovation, confirming the validity of the elements and dimensions of the Eco-Mi model and its potential for use. These pillars were shown as levers for the integration of eco-innovation in the petrochemical company, characterized by four main factors, associated to the four dimensions of the Eco-Mi model: sustainability as a success factor (strategic dimension); delegation and decentralization (structure dimension); empowerment (resources dimension); direct interactions with stakeholders (culture dimension). The representativeness of the case, due to the adequacy of the model to the circumstances in which eco-innovation practices occur and to the understanding of its specificities, can generate several possibilities of insights for the field of knowledge and for business practice.

It is concluded that the objectives of the research were achieved through the development and validation of an eco-innovation maturity model (Eco-Mi model), which systematizes the practices of eco-innovation—according to four organizational dimensions and levels of eco-innovation—and proposes a prescriptive method that supports the assessment of organizational performance and the holistic integration of eco-innovation practices in enterprises. The holistic approach and the prescriptive method for assessing and improving the organization can be highlighted as the originality of this research. As a scientific contribution, the proposition of a framework for eco-innovation management practices can be highlighted, as well as the characterization of the levels of maturity of each practice. As a practical contribution to the industrial environment, it is proposed: a simple and practical guide to the selection and global integration of eco-innovation practices in organizations; an instrument for evaluating organizational performance in eco-innovation; a case study to disseminate better industrial practices.

However, the model has some limitations. In order to increase the external validity of the model it is necessary that multiple studies be carried out in industries of different segments and of different maturity levels. Additionally, although the Eco-Mi Assessment Tool proposes the stage of organizational maturity through a quantitative weighting of the results, a greater validation of the stage depends on a qualitative analysis by a specialist in the area, in order to consider the aspects of the sector and the profile of the company. This qualitative analysis limits the use of the method by self-assessment, since the analysis requires knowledge regarding strategy, structure, resources, and organizational culture. These limitations of the Eco-Mi model show promising directions for future research and for the consolidation of the model and dissemination of knowledge in eco-innovation. Thus, some suggestions for continuity and unfolding of this research can be highlighted:

- The application of the model in industries of different segments and of different maturity levels, verifying the comprehensiveness of the Eco-Mi model;
- Research that correlates high levels of organizational maturity of eco-innovation (Levels 4 and 5 of the Eco-Mi model) with the results of innovation development with high sustainable potential;
- Development of tangible indicators for each Eco-Mi model eco-innovation practice, as well as its practical validation;
- Development of an electronic system that supports the Eco-Mi application method, making it viable to use the Internet, and the dissemination of the guide to eco-innovation practices.

Therefore, the field of knowledge still offers ample possibilities for further research and consolidation of the Eco-Mi eco-innovation maturity model.

**Author Contributions:** Conceptualization, A.X.; methodology, A.X.; writing—original draft preparation, A.X.; formal analysis, T.R.; resources: A.A.; writing—review and editing, T.R., L.L. and L.S.; validation, T.R. and A.A. All authors have read and agreed to the published version of the manuscript.

**Funding:** This research was funded by FAPERJ (process 200.163/2015), CAPES (process BEX 4873/14-9) and CNPq (process 142249/2013-0).

**Conflicts of Interest:** The authors declare no conflict of interest.

## Appendix A

The final version of the Guide to Eco-innovation Practices, presented as follows, portrays a classification and systematization of practices in the four organizational dimensions (Strategy, Structure, Resources, and Culture) and according to the proposed maturity levels.

**Table A1.** Strategy.

| Who We Are, Where We Are, and How We Can Achieve Maturity in Eco-Innovation | | | References |
|---|---|---|---|
| | Level | **Strategic Diagnosis** | |
| PA1 | 2 | The company assesses opportunities and risks in environmental legislation | Donaire (1995); Ormazabal and Sarriegi (2012) |
| PA2 | 2 | The company assesses its strengths and weaknesses for integrating environmental requirements into the product development process | Correa et al. (2008); Morrish et al. (2011) |
| PA3 | 3 | The company assesses environmental opportunities and risks in industry and competitors | Tidd, Bessant and Pavitt (1997); Donaire (1995) |
| PA4 | 3 | The company evaluates its strengths and weaknesses for sustainable innovations | Rodrigues (2006); Kruglianskas and Gomes (2011) |
| PA5 | 4 | The company assesses opportunities and risks in the level of awareness of consumers and society | Donaire (1995); Sanches (2000) |
| PA6 | 5 | Innovation strategies are part of a (systematic) process of open innovation (inbound and outbound) | Chesbrough (2003); Dahlander and Gann (2010) |
| PA7 | 5 | The company works in a collaborative network, where the process of identifying opportunities and risks is facilitated by a systemic process | Tachizawa and Andrade (2008); Manzini and Vezzoli (2002); Barbieri et al. (2010) |
| | | **Formulation of Strategies** | |
| PA8 | 3 | The company analyzes and implements socio-environmental strategies | Bonn and Fischer (2011); Coral (2002) |
| PA9 | 3 | The company incorporates the environmental dimension into strategic goals and indicators | Kaplan and Norton (2000); Coral (2002); Hockerts (2001); Campos and Selig (2011) |
| PA10 | 3 | The company incorporates the social dimension into strategic goals and indicators | Kaplan and Norton (2000); Coral (2002); Hockerts (2001); Campos and Selig (2011) |
| PA11 | 3 | Sustainability is integrated into the company mission | Coral (2002); Morrish et al. (2011) |
| PA12 | 4 | The company develops a strategic plan to assist decision making and management of the different sources (internal and external) of innovation | Porter (1980); Gomes (2009); Kaplan and Norton (2008) |
| PA13 | 4 | Economic, environmental, and social factors are fully incorporated into the company's innovation development processes | Coral (2002); Correa et al. (2008) |
| PA14 | 4 | Innovation strategies emerge from suppliers, customers, competitors, companies from other sectors, employees, institutes, and research centers | Kruglianskas and Gomes (2011) |
| PA15 | 4 | Strategies and operations are interconnected through a closed-loop management system—control system with an active feedback loop | Kaplan and Norton (2008) |
| PA16 | 4 | The company fully integrates the dimensions of sustainability into all strategic planning perspectives | Morrish et al. (2011); Coral (2002) |
| PA17 | 5 | The decision-making process can be carried out from the strategic level to the operational, from the operational to the strategic, and from the tactical to the strategic and operational, through a synergy that involves the whole organization | Mintzberg and Quinn (1991); Pigosso (2012); Gouvinhas et al. (2016) |
| | | **Monitoring and Control** | |
| PA18 | 2 | The company has an Environmental Management System | Courville (2004); Oliveira et al. (2014); Instituto Ethos (2000); Afnor (2003); British Standards Institution (1992; 1996); Canadian Standard Association (1993) |
| PA19 | 2 | The company develops sustainability reports to provide information regarding its social and environmental performance | Courville (2004); Oliveira et al. (2014); Instituto Ethos (2000) |
| PA20 | 2 | The company has a Social Responsibility Management System | Courville (2004); Oliveira et al. (2014); Instituto Ethos (2000); Canadian Standard Association |
| PA21 | 3 | The company has codes of conduct or sets of principles, with a normative orientation towards sustainability | Courville (2004); Oliveira et al. (2014); Instituto Ethos (2000) |
| PA22 | 3 | The company has an Integrated Management System, which integrates the processes of quality, health and safety, environmental management, and social responsibility | Instituto Ethos (2000); QSP (2003) |
| PA23 | 4 | The company uses indicators to measure performance in accordance with the Code of Conduct and with the objectives of the Management Systems | Courville (2004); Oliveira et al. (2014); Instituto Ethos (2000); GRI (2000;2008); Rees (1992); WEF (2002); Dow Jones (1999); BOVESPA (2005) |

**Table A2.** Structure.

| | How We Organize Ourselves to Implement Eco-Innovation Strategies | | References |
|---|---|---|---|
| | Level | **Organizational Architecture** | |
| PU1 | **2** | The company makes changes in its internal environment to suit the socio-environmental issues | Sanches (2000); Corazza (2003); De Oliveira (2004) |
| PU2 | **2** | The company develops systems and structures to support innovation without restricting it | Thompson et al. (2007); Leadbeater and Oakley (1999) |
| PU3 | **3** | The company incorporates functions and tasks of environmental management in the diverse routines and competence areas | Sanches (2000); Corazza (2003) |
| PU4 | **3** | The company uses intra and interdepartmental interaction mechanisms to foster the exchange of ideas and information (committees, discussion forums, thematic leaders) | Sanches (2000); Aligleri et al. (2009); Gardim et al. (2011); Silveira (2011) |
| PU5 | **3** | The company integrates aspects of sustainability through an environmental management department or through specific functions for this issue | Hunt e Auster (1995); Llerena (1996); Sanches (2000); Dyllick et al. (2000); Corazza (2003) |
| PU6 | **3** | The company develops an active work of (internal and external) communication about all the socio-environmental activities | Sanches (2000); Bommer and Jalajas (2004); Galvão (2014); Tidd & Bessant (2014) |
| PU7 | **4** | The company offers organizational spaces for free time to foster innovation (such as cafes, informal chat rooms, games room, gym) | Dougherty and Corse (1995); Bilton (2010); Tidd and Bessant (2014) |
| PU8 | **4** | The company establishes a time during the work day designated for the creation of socio-environmental projects by the employees | Bilton (2010); Tidd and Bessant (2014) |
| PU9 | **4** | The company works with an organic structure, more informal, flexible, and open to initiative | Burns and Stalker (1961); Porter (2006) |
| PU10 | **4** | The company provides mechanisms to enable stakeholder exchanges to integrate functional, technological, environmental, social, and cultural aspects | Sanches (2000); Husted and Allen (2001) |
| PU11 | **4** | The company seeks information on the skills of new players for potential partnerships | Gulati and Gargiolo (1999); Sanches (2000); Aligleri et al. (2009) |
| PU12 | **5** | The company works through a network structure, where partnerships ensure the generation of results for all members | Lipnack and Stamps (1994); Almeida (1995); Besanko et al. (1999); Wilkinson and Young (2002); Almeida et al. (2006) |
| | | **Leadership** | |
| PU13 | **3** | Senior managers are centrally involved in processes that define entrepreneurship for innovation | Ren and Guo (2011); Kuratko et al. (2014) |
| PU14 | **3** | Innovative initiatives also emerge from lower organizational levels and senior managers recognize the value of these ideas and direct them to the appropriate channels | Floyd and Lane (2000); Kuratko et al. (2014) |
| PU15 | **4** | Senior Managers demonstrate socio-environmental values leading eco-innovation internal initiatives and directly engaging in collaborative networks | Floyd and Lane (2000); Coral (2002); Hornsby et al. (2009) |
| PU16 | **5** | The company's leadership is recognized in the external environment as a reference in eco-innovation | Leifer et al. (2002); Coral (2002); Giddens (2003) |
| | | **Processes** | |
| PU17 | **2** | The company invests in new incremental organizational methods to face the challenges for sustainability | Zadec (2004); Galvão (2014) |
| PU18 | **2** | The company develops projects to reduce negative environmental impacts | Arnold and Hockerts (2011); Carrillo-Hermosilla et al. (2009) |
| PU19 | **2** | Environmental practices are considered (in the final stages) in the NPD (filters, effluent treatment, waste reduction) | Hauschild et al. (2005); Luttropp and Lagerstedt (2006) |
| PU20 | **3** | Sustainability is an explicit goal and an integrated success factor of the innovation process in the development of new products | Jones (2003); Siebenhüner and Arnold (2007) |
| PU21 | **3** | The company evaluates the eco-innovation process through specific indicators | Jones (2003); Pigosso (2012) |
| PU22 | **3** | The company uses combined ecodesign tools, contributing to the integration of environmental aspects into the NPD | Silveira (2006); Pigosso et al. (2013) |
| PU23 | **3** | Ecodesign practices are systematically incorporated into the development of products and processes from the early stages | Silveira (2006); Pigosso (2012) |
| PU24 | **3** | The company uses ecodesign tools that incorporate social sustainability criteria | Vallet et al. (2014) |
| PU25 | **3** | The implementation of ecodesign can be initiated by top management and also by designers and product developers (operational levels) | Pigosso (2012); Gouvinhas et al. (2016) |

**Table A3.** Resources.

| | How We Mobilize the Resources Needed to Achieve Eco-Innovation Strategies | | References |
|---|---|---|---|
| | Level | **Human Resources** | |
| PR1 | 2 | The company develops specific training programs to stimulate employees' creativeness | Kaur (2011) |
| PR2 | 2 | The company develops its employees to implement and operate an Environmental Management System | Jabbour et al. (2009) |
| PR3 | 2 | The company develops environmental training programs to all employees | Hunt and Auster (1995); Perron et al. (2006); Jabbour et al. (2009) |
| PR4 | 3 | The company develops environmental training programs to all outsourced employees | Hunt and Auster (1995); Jabbour et al. (2009) |
| PR5 | 3 | The company encourages the recruitment and development of creative, entrepreneurial, risk-prone, and environmentally minded people | Leifer et al. (2002); Jabbour et al. (2009); Silva et al. (2012) |
| PR6 | 3 | The company organizes workshops and lectures on themes related to innovation, sustainability, and its challenges, to promote collective awareness | Kruglianskas et al. (2009); Laville (2009) |
| PR7 | 3 | The company identifies the socio-environmental issues that concern intrapreneurs and establishes organizational objectives to respond to this | Almeida et al. (1993); Zadek (1998); Bateman and Snell (2002) |
| PR8 | 3 | The company has a policy of involvement with the community, respecting local customs and cultures and promoting social improvements | Tachizawa and Andrade (2006) |
| PR9 | 3 | The company announces its programs and its environmental policies in job fairs | Liebowtz (2010) |
| PR10 | 3 | The company uses a feedback system to evaluate and provide feedback to employees, particularly regarding environmental improvement efforts | Bateman and Snell (2002); Dutra (2002); Kaur (2011) |
| PR11 | 3 | The company organizes teams focused on recycling | Liebowtz (2010) |
| PR12 | 4 | Environmental management has a systemic approach that integrates sustainability at all organizational levels | Silveira (2006); Jabbour et al. (2010) |
| PR13 | 4 | Environmental requirements and targets are clear at all levels of the organization | Hunt and Auster (1995); Bateman and Snell (2002) |
| PR14 | 4 | The company leads/participates in initiatives and discussion groups with regulatory agents, with the government and also in local communities | Hunt and Auster (1995); Tachizawa and Andrade (2006) |
| PR15 | 4 | The company develops reward programs for innovative teams that promote ideas that contribute to improvement and eco-innovation | Galbraith (1995); Nolan and Croson (1996); Dutra (2002); Strachan et al. (2003); Massoud et al. (2008); Liebowtz (2010); Dutta (2012) |
| PR16 | 4 | The company empowers its employees as a way to increase commitment, improve decisions, innovation, and environmental awareness | Massoud et al. (2008); Kaur (2011) |
| PR17 | 4 | The company includes environmental management criteria in evaluating employee performance, enhancing the company's environmental culture | Bateman and Snell (2002); Renwick et al. (2008); Liebowtz (2010) |
| PR18 | 4 | The company promotes continuing environmental education programs to all employees, from upper management to the bottom of the pyramid | Tachizawa and Andrade (2006) |
| | | **Financial Resources** | |
| PR19 | 3 | The company has a fundraising strategy to guarantee greater autonomy to funders and guarantee their mission and values | Valarelli (1999); Leifer et al. (2002) |
| PR20 | 3 | Investments in RD&I (research, development, and innovation) focus on factors related to sustainability | Silveira (2006); Santos (2009) |
| PR21 | 4 | The company empowers the project manager or team member to raise funds and develop funding proposals for innovation | Leifer et al. (2002) |
| PR22 | 5 | The company is listed in Sustainable Stock Indices and it leverages itself to raise funds in the financial market | Santos (2009); Bovespa (2006) |
| PR23 | 5 | The data related to sustainability receive the same treatment (weight) as the financial data and both categories are audited with the same rigor | Silveira (2006) |
| | | **Infrastructure** | |
| PR24 | 2 | The company establishes adequate support for technological innovation, through the provision of technological infrastructure and training of employees | Plonski (2005); Wiig and Wood (1997); |
| PR25 | 2 | The technological structure consists of knowledge-oriented technologies as well as a set for computing and communication that accounts for social, environmental and economic performance and values | Davenport and Prusak (1998); Bachmann (1999); Silveira (2006) |
| PR26 | 2 | The company has a flexible organizational infrastructure that enables it to respond quickly to market and economic challenges | Stoeckicht and Soares (2010) |
| PR27 | 3 | The company uses new information technologies to reduce costs and to the collective use of knowledge, technology, productive and commercial means | Davenport and Prusak (1998); Bachmann (1999); Viaro (2011) |
| PR28 | 3 | The company uses new information technologies to integrate interests and for cooperation between customers, companies, and suppliers | Handy (1997); Olson (1999); Galina (2003); Campos (2011) |
| PR29 | 4 | The company has a space for ideation, as a way to promote the creation of new ideas, knowledge management, and communication | Davila et al. (2006) |
| PR30 | 5 | The company encourages the development of service and infrastructure platforms that support sustainable models | Tachizawa and Andrade (2006); Silveira (2006); Campos (2011) |
| | | **Relational Competences** | |
| PR31 | 3 | The company participates in collective instances (forums, councils, events) to search and disseminate environmental management knowledge | Paulino et al. (2005); Kotler and Keller (2012) |
| PR32 | 4 | The company uses partnerships and alliances as a source of information and learning, favored through shared RD&I (research, development, and innovation) | Hillestad et al. (2010); Wagner (2007); Zahra et al. (2007); Galvão (2014); Neutzling (2014) |
| PR33 | 4 | The management of external sources of information is an integral part of the company's technological strategy for innovation | Tigre (2006); Gomes (2009); Galvão (2014) |
| PR34 | 5 | The company works in a cooperation network (employees, technology institutes) to promote the development and diffusion of innovations | Hagedoorn (2002); Zahra et al. (2007); Hitt et al. (2008); Cousins and Menduc (2006) |
| PR35 | 5 | The company has the participation of the community in projects of common interest (soil conservation, rational use of resources, exploitation of raw material, etc.) | Tachizawa and Andrade (2008) |
| PR36 | 5 | The scientific and technological knowledge of the company is characterized by the close relationship between several actors (universities, government, companies, etc.) | Heringer (2011) |
| PR37 | 5 | The socio-environmental responsibility of the company is associated to the various forms of dialogue and interactions with stakeholders, guided by ethical, social, and environmental aspects | Galvão (2014) |

**Table A4.** Culture.

| | Level | How to Integrate Values Across the Organization to Create an Environment Conducive to Eco-Innovation | References |
|---|---|---|---|
| | | **Culture for Eco-Innovation** | |
| PC1 | 2 | Company norms are concrete expressions of their values and incorporate the expectations of action in the organization | Katz and Kahn (1978); Foguel and Souza (1995); Baker et al. (2014) |
| PC2 | 2 | Company norms encourage people to take action to achieve innovation goals | Baker e Sinkula (2007); Colbert et al. (2008); Green et al. (2008); López and Cuervo-Arango (2008) |
| PC3 | 3 | The company establishes at all levels a strategy that values the acquisition, creation, accumulation, protection, and exploitation of knowledge | Sluis (2004); Baker et al. (2014) |
| PC4 | 4 | The company encourages entrepreneurial behavior, through the valorization of ideation and recognition of efforts | Volpato and Cimbalista (2002); Sluis (2004) |
| PC5 | 4 | The company promotes changes in the routines, to avoid the inflexibility of previous knowledge and to increase the capacity of innovation | Arruda et al. (2009); Yang et al. (2014) |
| PC6 | 4 | The company encourages values such as solidarity, equality, partnership, and cooperation | UNESCO (2005); Azevedo (2013) |
| | | **Organizational Climate** | |
| PC7 | 2 | The company encourages cooperation among members through work and team recognition | Dejours (1993); Volpato and Cimbalista (2002); Gagné and Deci (2005) |
| PC8 | 2 | The company integrates multidisciplinary competences into teams to perform complex and significant tasks | Büschgens et al. (2013) |
| PC9 | 2 | The company encourages the improvement of eco-innovation leadership, such as training, coaching programs, responsibility taking skills, etc. | Tidd and Bessant (2014) |
| PC10 | 3 | The remuneration policies, procedures, and systems reflect ethical organizational values, encouraging ethical behavior | Barnett and Vaicys (2000) |
| PC11 | 3 | The company encourages the members' autonomy and independence to perform creative and challenging tasks | Greenberg (1994); Alencar (1996); Langfred and Moye, 2004 |
| PC12 | 4 | The company promotes open, direct, and collaborative communication among all members, without repressing initiatives, opinions, and ideas | Alencar (1996); Crespo and Wechsler (2000); Tidd and Bessant (2014) |
| PC13 | 4 | The company establishes time and space for the promotion of creativity and innovation | Crespo and Wechsler (2000); Tidd and Bessant (2014) |
| PC14 | 5 | The company encourages proactivity by encouraging employees and stakeholders to acquire and share knowledge and make decisions | Greenberg (1994); Langfred and Moye (2004); Tidd and Bessant (2014) |
| | | **Organizational Learning** | |
| PC15 | 2 | The company develops informal electronic channels to support the exchange and sharing of technical-scientific information (Facebook, networks, blogs) | Vital (2006); Rizova (2006) |
| PC16 | 2 | The company has metrics to understand the key learning types for the company, based on innovation and sustainability criteria | Silveira (2006) |
| PC17 | 3 | The company promotes broad participation in knowledge generation and change, making continuous improvement efforts in this regard | Levine (2001); Garvin (1998) |
| PC18 | 3 | The company has metrics to understand the correct way to conduct the learning process, considering the specific social and cultural realities of the organization | Silveira (2006); Albuquerque (2011) |
| PC19 | 4 | The company encourages and supports the development and evolution of communities of practice | Gardim et al. (2011) |
| PC20 | 4 | The company encourages suppliers to also include social environmental concerns in their processes and procedures | Gouvinhas et al. (2016) |
| PC21 | 4 | Socio-environmental performance is used as a key criterion for choosing the company's suppliers | Gouvinhas et al. (2016) |
| PC22 | 4 | The company conducts direct consultation of workers' opinions on environmental improvement initiatives in development processes | Boiral (2002) |
| PC23 | 5 | The company promotes various forms of dialogue and interactions with stakeholders guided by ethical, social, and environmental aspects | Aligleri et al. (2009); Galvão (2014) |
| PC24 | 5 | The company is part of a collective learning process where the roles of each member are discussed according to the experiences and objectives of sustainability | Manzini and Vezzoli (2002) |
| PC25 | 5 | The company "educates" its clients on the importance of considering socio-environmental aspects during their buying decisions | Gouvinhas et al. (2016) |
| PC26 | 5 | The company conveys customer feedback for the entire value chain, involving a complete loop of information exchange | Gouvinhas et al. (2016) |
| PC27 | 5 | The company works in an integrated network of sustainable companies, where there is a constant exchange of experiences and encouragement to business partners for sustainability | Terra and Gordon (2002); Bismarchi (2011); Gouvinhas et al. (2016) |

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
