# Peer review of "Eco-Innovation Maturity Model: A Framework to Support the Evolution of Eco-Innovation Integration in Companies"

_sustainability, doi:10.3390/su12093773_

Round 1

Reviewer 1 Report

While the manuscript has several interesting finings, its scientific soundness remains unclear at many instances. In the current version, it is a rather unscientific appraoch with interesting results. Please see several comments below that must be adressed:

The introduction is like a combined background section and definitions. However, what the article wants to contribute exactly, remains unclear and is scattered in a quite long text. I suggest to divide the introduction into a much shorter, focused one and a background section. The literature review remains unclear. How was literature searched for, selected, and evaluated. Methodological references and much more information is required for this in academic journals. Background information is missing: Where do the 142 practices of eco-innovation come from? From the literature review? Please check your mauscript carefully since information seems to be missing here. The same is true for the stages and categories derived: Did the authors invent those, derive from literature, ...? From section 3.3 on, the article turns entirely away from references and academic rigor. While the content is interesting, it seems like a practice report. In section 3.4, we don't get any method for the interviews, how the interviewees were selected, etc. The case study lacks background information: Where was it done, how is it generalizable, ... The discussion does rather sum-up than discuss the paper with extant literature. The conclusion does not derive clear implications for theory, practice, but is also rather a sum-up.

Overall, this is like a report from practice across many parts of the manuscript, but not a scientific paper. I suggest the authors to rewrite the paper according to academic standards, as given above, or to reposition it as a practice report paper, not an academic article. There are possibilities such as "Case Report" for the journal Sustainability.

For an academic article, a more scientific writing style is required.

Further, only very few references from the journal Sustainability are cited.

Author Response

Dear Reviewer 1

I am delighted to inform you that the manuscript entitled “Eco-innovation maturity model: a framework to support the evolution of eco-innovation integration in companies" has been revised and adjusted, considering the suggestions included in the reports.

The authors are grateful for the reviewer’s valuable comments and suggestions, which contributed to improve the quality of the article. The amendments implemented are detailed below.

I’m looking forward to receiving your feedback in due course.

Sincerely, Corresponding Author

Point 1: The introduction is like a combined background section and definitions. However, what the article wants to contribute exactly, remains unclear and is scattered in a quite long text. I suggest to divide the introduction into a much shorter, focused one and a background section.

Response 1: Amendments implemented. The introduction was improved (lines 40-45; 72-75; 79-82; 85-88), through the highlight of the originality (lines 102-103) and the theoretical and practical contributions of the article (line 110-115). A Background theory field was created (pages 4-6), containing theoretical definitions - presented earlier in the introduction (lines 123-168) -, the theoretical foundation on the dimensions of innovation management (lines 169-194) and the state of the art (lines 195-236).

Point 2: The literature review remains unclear. How was literature searched for, selected, and evaluated. 

Response 2: Amendments implemented. The section Materials and Method was improved, in the sense that detailed explanations of the literature review procedure (lines 265-287) have been added.

Point 3: Further, only very few references from the journal Sustainability are cited.

Response 3: Amendments implemented. References from the journal Sustainability have been included (lines 39, 46, 56, Table 1 - related to lack of culture, line 137).

Point 4: Methodological references and much more information is required for this in academic journals.

Response 4: Amendments implemented. The section Materials and Method was improved, with the addition of references and detailed explanations of the methods and procedures used in the article (lines 255-260;265-291;301-305;312-316;317-333).

Point 5: Background information is missing: Where do the 142 practices of eco-innovation come from? From the literature review? Please check your mauscript carefully since information seems to be missing here.

Response 5: Amendments implemented. The section Materials and Method was improved, in the sense that detailed explanations of the literature review procedure (lines 265-287) have been added.

Point 6: The same is true for the stages and categories derived: Did the authors invent those, derive from literature, ...?

Response 6: Amendments implemented. A section on Background Theory was created, which contributes with detailed explanations and references about the dimensions of innovation management (lines 169-194). The Materials and Method section also presents this explanation briefly (lines 255-260). Section 4.2 has been improved, considering the addition of references regarding the maturity stage procedure (lines 397-402). The entire procedure for developing the model's maturity levels can be seen in the following paragraphs (lines 403-428).

Point 7: From section 3.3 on, the article turns entirely away from references and academic rigor.

Response 7: Amendments implemented. This section (4.3) was improved, through the addition of references (lines 435-439;468-472;493;534;550) and detailed explanations of the used procedures (lines 538-540;553-557;561-562).

Point 8: In section 3.4, we don't get any method for the interviews, how the interviewees were selected, etc.

Response 8: Amendments implemented. The section Materials and Method was improved, now containing detailed explanations about the Delphi method (lines 288-291, 301-305). The section 4.4 was also improved, with further references and explanations (lines 570;573;578;583-584;588;590;592).

Point 9: The case study lacks background information: Where was it done, how is it generalizable, ...

Response 9: Amendments implemented. The section Materials and Method was improved, now containing detailed explanations about the case study procedure (lines 306-333). The section 5 was also improved, with the addition of a summary description of the company (lines 627-654).  An explanation regarding the generalization of the case was added in section 5.4.2 (lines 849-854).

Point 10: The discussion does rather sum-up than discuss the paper with extant literature.

Response 10: Amendments implemented. The section 5.4 was improved, with the addition of references and explanations (709; 722-723; 732; 733-734; 736; 743-745; 758; 760; 767; 771; 772; 775; 777; 783; 795; 798; 803; 807; 825; 830; 832). An explanation regarding the generalization of the case was added in section 5.4.2 (lines 849-850).

Point 11: The conclusion does not derive clear implications for theory, practice, but is also rather a sum-up.

Response 11: Amendments implemented. The conclusion was improved, through the highlight of the representativeness of the case (lines 941-943), the originality (lines 908-910) and the theoretical and practical contributions of the article (lines 915-922).

Point 12: Extensive editing of English language and style required.

Response 12: Amendments implemented. The article was completely revised, with improvements in all topics.

Reviewer 2 Report

The Authors proposed and improved the model of the eco-innovation through expert evaluation using the Delphi Method, which allowed better accuracy. They confirmed the research hypothesis and, therefore, the validity of the Eco-Mi Model as support for the integration on of eco-innovation in organizations and as a reference for that field of knowledge.

This qualitative analysis restrict the use of the method by self-assessment, since the analysis requires  strategy, structure, resources and organizational culture. THe presented by the authors limitations of the Eco-Mi Model show directions for future research on this topic and dissemination of knowledge in eco-innovation. Some suggestions for continuity of this research were formulated by the authors as well.

The authors should clarify which ecoinnovation definitions they applied.

There is a mistake in numeration: Figure 41 - Coding of the practices of the Eco-Mi model. Some figures eg. number 3 are not visisble, due to small font.

Author Response

Dear Reviewer 2

I am delighted to inform you that the manuscript entitled “Eco-innovation maturity model: a framework to support the evolution of eco-innovation integration in companies" has been revised and adjusted, considering the suggestions included in the reports.

The authors are grateful for the reviewer’s valuable comments and suggestions, which contributed to improve the quality of the article. The amendments implemented are detailed below.

I’m looking forward to receiving your feedback in due course.

Sincerely, Corresponding Author

Point 1: The authors should clarify which eco-innovation definitions they applied.

Response 1: Amendments implemented. A paragraph has been added on the subject (lines 38-44).

Point 2: There is a mistake in numeration: Figure 41 - Coding of the practices of the Eco-Mi model.

Response 2: Amendments implemented.

Point 3: Some figures eg. number 3 are not visisble, due to small font.

Response 3: Amendments implemented. All figures and tables have been revised and improved.

Other points

Point 4: The introduction can be improved.

Response 4: The introduction was improved (lines 40-45; 72-75; 79-82; 85-88), through the highlight of the originality (lines 102-103) and the theoretical and practical contributions of the article (line 110-115). A Background theory field was created (pages 4-6), containing theoretical definitions - presented earlier in the introduction (lines 123-168) -, the theoretical foundation on the dimensions of innovation management (lines 169-194) and the state of the art (lines 195-236).

Point 5: The methods can be improved

Response 5: The section Materials and Method was improved, with the addition of references and detailed explanations of the methods and procedures used in the article (lines 255-260;265-291;301-305;312-316;317-333).

Point 6: The results can be improved.

Response 6: Amendments implemented. The section 5.4 was improved, with the addition of references and explanations (709; 722-723; 732; 733-734; 736; 743-745; 758; 760; 767; 771; 772; 775; 777; 783; 795; 798; 803; 807; 825; 830; 832). An explanation regarding the generalization of the case was added in section 5.4.2 (lines 849-850).

Point 7: The conclusions can be improved.

Response 7: The conclusion was improved, through the highlight of the representativeness of the case (lines 941-943), the originality (lines 908-910) and the theoretical and practical contributions of the article (lines 915-922).

Point 8: English language and style are fine/minor spell check required

Response 8: Amendments implemented. The article was completely revised, with improvements in all topics.

Reviewer 3 Report

Dear authors, I found very interesting your study. The variables that you have been used in the analysis are well explained. But after reading the work, I had found the doctoral thesis from which it came, published on the internet with the same content and the same figures and tables.

This situation should be reviewed and assessed by the journal.

The full content of the article is published on the Internet with the doctoral thesis:

Xavier, A. F. (2017). PROPOSTA DE UM MODELO DE MATURIDADE PARA AVALIAÇÃO DAS PRÁTICAS DE ECO-INOVAÇÃO NAS ORGANIZAÇÕES: ECO-MI (Doctoral dissertation, Universidade Federal do Rio de Janeiro).

Comment 1. The figures and tables of the whole document are published in the previous doctoral thesis document according to:

Figure 1. In the theses: Figure 1, page 12 Figure 2. In the theses: Quadro 1(page 11) and Figure 5 (page 21) Figure 3. In the theses: Figure 26, page 131. Similar in page 31. Figure 4. In the theses: Figure 30, page 127 Figure 5. In the theses: Figure 31, page 139 Figure 6. In the theses: Figure 48, page 159 Figure 7. In the theses: Figure 33, page 144 Figure 8. In the theses: Figure 34, page 145 Figure 9. In the theses: Figure 37, page 148 Figure 10. In the theses: Figure 60, page 206 Table 1, is contained in the text. Table 2. In the theses: Table 17, page 160 Table 3. In the theses: Table 21, page 172

Comment 2. Lines from 41 to 43 contain the same information than lines 74 to 75.

Comment 3. In line 59 the reference is not well cited. Instead write the citation as follows: [7-10].

Comment 4. Same for line 99. Should be [20-24].

Comment 5. In line 59 the reference is not well cited. Instead write the citation as follows: [7-10].

Comment 6. Tables and figures should be in journal format.

For example: Change Table 1 - Organize.... to Table 1. Organize….

In the case of the figures the situation is the same.

Comment 7. Figure 41 (line 257) should be Figure 4.

Comment 8. Figure 10 (line 502) appears with ":" should be corrected to the format of the journal.

 Comment 9. The titles of lines 568, 590, 609 and 622 should be reviewed and agreed with the journal. I don't think it's appropriate to put the items in that format.

Comment 10. The work contains references until 2017 because its content belongs to the referenced thesis that dates from April 2017. Except for three references that have been added from 2019.

Author Response

Dear Reviewer 3

I am delighted to inform you that the manuscript entitled “Eco-innovation maturity model: a framework to support the evolution of eco-innovation integration in companies" has been revised and adjusted, considering the suggestions included in the reports.

The authors are grateful for the reviewer’s valuable comments and suggestions, which contributed to improve the quality of the article. The amendments implemented are detailed below.

I’m looking forward to receiving your feedback in due course.

Sincerely, Corresponding Author

Point 1: The figures and tables of the whole document are published in the previous doctoral thesis document according to: Figure 1. In the theses: Figure 1, page 12 Figure 2. In the theses: Quadro 1(page 11) and Figure 5 (page 21) Figure 3. In the theses: Figure 26, page 131. Similar in page 31. Figure 4. In the theses: Figure 30, page 127 Figure 5. In the theses: Figure 31, page 139 Figure 6. In the theses: Figure 48, page 159 Figure 7. In the theses: Figure 33, page 144 Figure 8. In the theses: Figure 34, page 145 Figure 9. In the theses: Figure 37, page 148 Figure 10. In the theses: Figure 60, page 206 Table 1, is contained in the text. Table 2. In the theses: Table 17, page 160 Table 3. In the theses: Table 21, page 172.

Response 1: The Assistant Editor of the Journal contacted me about this subject. I informed her that the images and tables are directly linked to the thesis, but that they were not submitted to any peer review and were not published in any other journal. The Assistant Editor did not make any comments or requests in this regard. I am available to make adjustments to all figures and tables in the article if the journal and the reviewers believe it is necessary to differentiate from those published in the thesis.

Point 2: Lines from 41 to 43 contain the same information than lines 74 to 75.

Response 2: Amendments implemented. The sentence has been changed (line 75).

Point 3: In line 59 the reference is not well cited. Instead write the citation as follows: [7-10].

Response 3: Amendments implemented.

Point 4: Same for line 99. Should be [20-24].

Response 4: Amendments implemented.

Point 5: In line 59 the reference is not well cited. Instead write the citation as follows: [7-10].

Response 5: Amendments implemented.

Point 6: Tables and figures should be in journal format. For example: Change Table 1 - Organize.... to Table 1. Organize….

In the case of the figures the situation is the same.

Response 6: Amendments implemented.

Point 7: Figure 41 (line 257) should be Figure 4

Response 7: Amendments implemented.

Point 8: Figure 10 (line 502) appears with ":" should be corrected to the format of the journal.

Response 8: Amendments implemented.

Point 9: The titles of lines 568, 590, 609 and 622 should be reviewed and agreed with the journal. I don't think it's appropriate to put the items in that format.

Response 9: Amendments implemented.

Point 10: The work contains references until 2017 because its content belongs to the referenced thesis that dates from April 2017. Except for three references that have been added from 2019.

Response 10: Amendments implemented. New references have been included (lines 39, 46, 56, Table 1 - related to lack of culture, line 137).

Other points

Point 4: The introduction must be improved.

Response 4: The introduction was improved (lines 40-45; 72-75; 79-82; 85-88), through the highlight of the originality (lines 102-103) and the theoretical and practical contributions of the article (line 110-115). A Background theory field was created (pages 4-6), containing theoretical definitions - presented earlier in the introduction (lines 123-168) -, the theoretical foundation on the dimensions of innovation management (lines 169-194) and the state of the art (lines 195-236).

Point 5: The methods can be improved

Response 5: The section Materials and Method was improved, with the addition of references and detailed explanations of the methods and procedures used in the article (lines 255-260;265-291;301-305;312-316;317-333).

Point 6: The results must be improved.

Response 6: Amendments implemented. The section 5.4 was improved, with the addition of references and explanations (709; 722-723; 732; 733-734; 736; 743-745; 758; 760; 767; 771; 772; 775; 777; 783; 795; 798; 803; 807; 825; 830; 832). An explanation regarding the generalization of the case was added in section 5.4.2 (lines 849-850).

Point 7: The conclusions must be improved.

Response 7: The conclusion was improved, through the highlight of the representativeness of the case (lines 941-943), the originality (lines 908-910) and the theoretical and practical contributions of the article (lines 915-922).

Round 2

Reviewer 1 Report

Dear authors,

thank you for your revisions. I have one comment left:

The model and case study is considering the petrochemical industry. It would be interesting to discuss how this industry compares to other ones. Please see two papers with industry sector comparisons from Sustainability below that could be used to discuss this:

Müller, J.M. Antecedents to Digital Platform Usage in Industry 4.0 by Established Manufacturers. Sustainability 2019, 11, 1121.

Cioca, L.-I.; Ivascu, L.; Rada, E.C.; Torretta, V.; Ionescu, G. Sustainable Development and Technological Impact on CO2 Reducing Conditions in Romania. Sustainability 2015, 7, 1637-1650.

Author Response

Dear Reviewer 1

I am delighted to inform you that the manuscript entitled “Eco-innovation maturity model: a framework to support the evolution of eco-innovation integration in companies" has been revised and adjusted, considering the suggestions included in the reports.

The authors are grateful for the reviewer’s valuable comments and suggestions, which contributed to improve the quality of the article. The amendments implemented are detailed below.

I’m looking forward to receiving your feedback in due course.

Sincerely, Corresponding Author

Point 1: The model and case study is considering the petrochemical industry. It would be interesting to discuss how this industry compares to other ones. Please see two papers with industry sector comparisons from Sustainability below that could be used to discuss this:

Müller, J.M. Antecedents to Digital Platform Usage in Industry 4.0 by Established Manufacturers. Sustainability 2019, 11, 1121.

Cioca, L.-I.; Ivascu, L.; Rada, E.C.; Torretta, V.; Ionescu, G. Sustainable Development and Technological Impact on CO2 Reducing Conditions in Romania. Sustainability 2015, 7, 1637-1650.

 Response 1: Amendments implemented.  A section for discussion on the Petrochemical Industry was created (section 5.1) (lines 627-657).

Point 2: English language and style are fine/minor spell check required.

Response 2: Amendments implemented. The article was completely revised, with improvements.

Reviewer 3 Report

Dear authors,

The article has improved substantially but there are still some small details to be improved:

  • line 347: the figure is cited (Fig.3) and should be Figure 3 to maintain the format of the rest of them.
  • line 508: the same applies to the figure 9 (delete Fig.9)
  • Figure 8 is not mentioned in the text.
  • Line 265: should be removed [AX1]
  • Line 485: prevent the figure footer being moved to another page.

Kind regards!

Author Response

Dear Reviewer 3

I am delighted to inform you that the manuscript entitled “Eco-innovation maturity model: a framework to support the evolution of eco-innovation integration in companies" has been revised and adjusted, considering the suggestions included in the reports.

The authors are grateful for the reviewer’s valuable comments and suggestions, which contributed to improve the quality of the article. The amendments implemented are detailed below.

I’m looking forward to receiving your feedback in due course.

Sincerely, Corresponding Author

Point 1: line 347: the figure is cited (Fig.3) and should be Figure 3 to maintain the format of the rest of them.

Response 1: Amendments implemented.

Point 2: line 508: the same applies to the figure 9 (delete Fig.9)

Response 2: Amendments implemented.

Point 3: Figure 8 is not mentioned in the text.

Response 3: Amendments implemented.

Point 4: Line 265: should be removed [AX1]

Response 4: Amendments implemented.

Point 5: Line 485: prevent the figure footer being moved to another page.

Response 5: Amendments implemented.